# Action Gaps and Advantages in Continuous-Time Distributional Reinforcement Learning

**Harley Wiltzer**[*]
Mila–Québec AI Institute
McGill University

**Marc G. Bellemare**[†]
Mila–Québec AI Institute
McGill University

**David Meger**
McGill University

**Patrick Shafto**
Rutgers University–Newark

**Yash Jhaveri**[*]
Rutgers University–Newark

## Abstract

When decisions are made at high frequency, traditional reinforcement learning (RL) methods struggle to accurately estimate action values. In turn, their performance is inconsistent and often poor. Whether the performance of distributional RL (DRL) agents suffers similarly, however, is unknown. In this work, we establish that DRL agents *are* sensitive to the decision frequency. We prove that action-conditioned return distributions collapse to their underlying policy's return distribution as the decision frequency increases. We quantify the rate of collapse of these return distributions and exhibit that their statistics collapse at different rates. Moreover, we define distributional perspectives on action gaps and advantages. In particular, we introduce the *superiority* as a probabilistic generalization of the advantage— the core object of approaches to mitigating performance issues in high-frequency value-based RL. In addition, we build a superiority-based DRL algorithm. Through simulations in an option-trading domain, we validate that proper modeling of the superiority distribution produces improved controllers at high decision frequencies.

## 1 Introduction

In many real-time deployments of reinforcement learning (RL)—quantitative finance, robotics, and autonomous driving, for instance—the state of the environment evolves continuously in time, but policies make decisions at discrete timesteps ($h$ units of time apart) [28]. In such systems, the performance of value-based agents is sensitive to the frequency $\omega := 1/h$ with which actions are taken. In particular, action values become indistinguishable as the time between actions decreases. In turn, in high-frequency settings, Baird demonstrated that action value estimates are susceptible to noise and approximation error [20]. Moreover, Tallec et al. exhibited that the performance of popular deep $Q$-learning agents is inconsistent and often poor [34].

In order to remedy this sensitivity, Baird proposed the advantage function and advantage-based variants of $Q$-learning, Advantage Updating (AU) [20] and Advantage Learning (AL) [2]. Unlike action values, advantages (appropriately rescaled) do not become indistinguishable as decision frequency increases. As a result, Baird, in [20, 2], demonstrated that advantage-based agents can learn faster and be more resilient to noise than their action value-based counterparts. Furthermore, Tallec et al., in [34], exhibited that their extension of AU, Deep Advantage Updating (DAU), works efficiently over a wide range of timesteps and environments, unlike standard deep $Q$-learning approaches.

While advantage-based approaches to RL have demonstrated robustness to decision frequency, in this work, we establish that they are nevertheless sensitive to the frequency with which actions

---

[*]Equal contribution. Correspondence to `harley.wiltzer@mail.mcgill.ca`.
[†]CIFAR AI Chair.

are taken. This discovery arises as we answer the question: to what extent is the performance of distributional RL (DRL) agents sensitive to decision frequency? To this end, we build theory within the formalism of continuous-time RL where environmental dynamics are governed by SDEs, as in [25]. Additionally, we validate our theory empirically through simulations. Specifically, we make the following four contributions:

**Distributional Action Gap.** First, we extend notions of action gap to the realm of DRL. Precisely, we consider the minimal distance between pairs of action-conditioned distributions under metrics on the space of probability measures on $\mathbb{R}$. We observe that some metrics are viable for this extension, while others are not. This formalism sets the stage for analyzing the influence of individual actions as well as decision frequency on, for example, an agent's return distributions.

**Collapse of Distributional Control at High Frequency.** Second, we establish tight bounds on the distributional action gaps of *h-dependent action-conditioned return distributions*—return distributions induced by applying a specific initial action for $h$ units of time. We prove that these distributional action gaps not only collapse, as $h$ tends to zero, but do so at a *slower rate* than action-value gaps. On one hand, therefore, distributional $Q$-learning algorithms are susceptible to the same failures as $Q$-learning in continuous-time RL. On the other hand, however, remedies to these failures transliterated to distributional $Q$-learning algorithms are unlikely to succeed, because the means of these return distributions collapse *faster* than their other statistics.

**Distributional Superiority.** Third, we propose an axiomatic construction of a distributional analogue of the advantage, which we call the *superiority*. Leveraging our analysis of $h$-dependent action-conditioned returns and their distributional action gaps, we present a frequency-scaled superiority distribution that enables greedy action selection at any fixed decision frequency.

**A Distributional Action Gap-Preserving Algorithm.** Fourth, we propose an algorithm that learns the superiority distribution from data. Empirically, we demonstrate that our algorithm maintains the ability to perform policy optimization at high frequencies more reliably than existing methods.

## 2 Setting

**Notation:** Spaces will either be subsets of Euclidean space or discrete. Measurability, in the former case, will be with respect to the Borel sigma algebra; in the latter case, it will be with respect to the power set. The set of probability measures over a space $\mathsf{Y}$ will be denoted by $\mathscr{P}(\mathsf{Y})$. Functions on spaces are assumed to be measurable. For $f : \mathsf{Y} \to \mathsf{Z}$ and $\mu \in \mathscr{P}(\mathsf{Y})$, the *push forward* of $\mu$ through/by $f$, $f_\# \mu \in \mathscr{P}(\mathsf{Z})$, is defined by $f_\# \mu := \mu \circ f^{-1}$. For a random variable $X$, defined implicitly on some probability space $(\Omega, \mathcal{F}, \mathbf{P})$, we write $\mathrm{law}(X) := X_\# \mathbf{P}$ to denote the law of $X$; the notation $X =_{\mathrm{law}} Y$ is shorthand for $\mathrm{law}(X) = \mathrm{law}(Y)$. For any $\mu \in \mathscr{P}(\mathbb{R})$, the *quantile function of $\mu$*, $F_\mu^{-1}$, is defined by $F_\mu^{-1}(\tau) := \inf_z \{F_\mu(z) \geq \tau\}$, where $F_\mu(z)$ is the CDF of $\mu$.

### 2.1 Continuous-Time RL

Here we give a brief introduction to the technical aspects of continuous-time RL, à la [25]. We provide additional exposition and references in Appendix A. For any reader looking to defer some of this technical introduction, we summarize the core objects of interest at the end of Section 2.1.1.

#### 2.1.1 MDPs

Continuous-time Markov Decision Processes (MDPs) are defined by three spaces and four measurable functions: a time interval $\mathsf{T} := [0, T]$ with $T \in (0, \infty)$ or $\mathsf{T} := [0, T)$ with $T = \infty$, a state space $\mathsf{X} \subset \mathbb{R}^n$, an action space $\mathsf{A}$, a drift $b : \mathsf{T} \times \mathsf{X} \times \mathsf{A} \to \mathbb{R}^n$, a diffusion $\sigma : \mathsf{T} \times \mathsf{X} \times \mathsf{A} \to \mathbb{R}^{n \times n}$, a reward $r : \mathsf{T} \times \mathsf{X} \to \mathbb{R}$, and a terminal reward $f : \mathsf{X} \to \mathbb{R}$.[3] The pair $(b, \sigma)$ govern the environment's dynamics by a family of SDEs parameterized by $a \in \mathsf{A}$,

$$\mathrm{d}X_t^a = b(t, X_t^a, a)\, \mathrm{d}t + \sigma(t, X_t^a, a)\, \mathrm{d}B_t. \tag{2.1}$$

Here $(B_t)_{t \geq 0}$ is an $n$-dimensional Brownian motion. In turn, any solution to (2.1) collects the state paths of an agent that chooses action $a$ at every time, regardless of the state they are in.

As is done in discrete-time RL, an agent might consider the induced Markov Reward Process (MRP) derived from a policy $\pi : \mathsf{T} \times \mathsf{X} \to \mathscr{P}(\mathsf{A})$.[4] The dynamics of a policy-induced MRP (with policy $\pi$)

---

[3] We work with action-independent rewards. This is the norm in continuous-time DRL.

[4] We assume that $(t, x) \mapsto \pi(E \mid t, x)$ is measurable for every $E \subset \mathsf{A}$ and every $(t, x) \in \mathsf{T} \times \mathsf{X}$.

are governed by the SDE

$$\mathrm{d}X_t^\pi = b^\pi(t, X_t^\pi)\,\mathrm{d}t + \sigma^\pi(t, X_t^\pi)\,\mathrm{d}B_t. \tag{2.2}$$

Here, following [38, 15, 16], the policy-averaged coefficients $b^\pi$ and $\sigma^\pi$ are defined by

$$b^\pi(t, x) := \int_{\mathsf{A}} b(t, x, a)\,\pi(\mathrm{d}a \,|\, t, x) \quad \text{and} \quad \sigma^\pi(t, x) := \left( \int_{\mathsf{A}} \sigma\sigma^\top(t, x, a)\,\pi(\mathrm{d}a \,|\, t, x) \right)^{1/2}. \tag{2.3}$$

Thus, solutions to (2.2) collect the paths of an agent following policy $\pi$.

A class of policies central to our study is those that fix an action $a$ from some time $t$ for a given *persistence horizon* $h$.

**Definition 2.1.** *Given $h > 0$ and $a \in \mathsf{A}$, a policy $\pi$ is said to be $(h, a)$-persistent at time $t \in \mathsf{T}$ if $\pi(\cdot \,|\, s, y) = \delta_a$ for all $(s, y) \in [t, t+h) \times \mathsf{X}$.*

In particular, given a policy $\pi$, we will consider $(h, a)$-*persistent modifications of* $\pi$: for $t \in \mathsf{T}$,

$$\pi|_{h,a,t}(\cdot \,|\, s, y) := \begin{cases} \delta_a & \text{if } s \in [t, t+h) \\ \pi(\cdot \,|\, s, y) & \text{if } s \notin [t, t+h). \end{cases}$$

These policies will help us understand the influence of taking actions relative to others as well as to those taken by $\pi$. We assume $h$ is small enough so that $t + h \in \mathsf{T}$.

In order to guarantee the global-in-time existence and uniqueness of solutions to our SDEs (2.1) and (2.2), we make two sets of assumptions.

**Assumption 2.2.** *The functions $b$ and $\sigma$ have linear growth and are Lipschitz in state, uniformly in time and action: a finite, positive constants $C_{2.2}$ exists such that*

$$\sup_{t,a} |b(t, x, a)| + \sup_{t,a} |\sigma(t, x, a)| \leq C_{2.2}(1 + |x|) \quad \forall x \in \mathsf{X}; \quad \text{and}$$

$$\sup_{t,a} |b(t, x, a) - b(t, y, a)| + \sup_{t,a} |\sigma(t, x, a) - \sigma(t, y, a)| \leq C_{2.2}|x - y| \quad \forall x, y \in \mathsf{X}.$$

**Assumption 2.3.** *The averaged coefficient functions $b^\pi$ and $\sigma^\pi$ are Lipschitz in state, uniformly in time: a finite, positive constant $C_{2.3}$ exists such that*

$$\sup_t |b^\pi(t, x) - b^\pi(t, y)| + \sup_t |\sigma^\pi(t, x) - \sigma^\pi(t, y)| \leq C_{2.3}|x - y| \quad \forall x, y \in \mathsf{X}.$$

These assumptions are standard in the analysis of continuous-time RL, optimal control, and SDEs [12, 26, 38, 16, 42]. Since $\pi$ is a function of state, we note that Assumption 2.3 is not a direct consequence of Assumption 2.2. The coefficients $b^\pi$ and $\sigma^\pi$ satisfy the conditions of Assumption 2.3 provided $b$ and $\sigma$ satisfy some (also standard) additional regularity conditions and $\pi$ satisfies some regularity conditions. These technical details are discussed in Appendix A.2.

In summary, in continuous-time RL, there are three stochastic processes of interest: $(X_s^\bullet)_{s \geq t}$ with $\bullet \in \{a, \pi, \pi|_{h,a,t}\}$, all beginning at some time $t$. These processes collect the state paths of an agent in one of three scenarios: 1. choosing action $a$ at every state and time; 2. following a policy $\pi$; or 3. choosing $a$ at every state and time for the first $h$ units of time and following $\pi$ thereafter.

### 2.1.2 Value Functions and their Distributions

Given a policy-induced state process $(X_s^\pi)_{s \geq t}$, the (discounted) random *return* $G^\pi(t, x)$ earned by $\pi$ starting from state $x \in \mathsf{X}$ at time $t \in \mathsf{T}$ is defined [41] by

$$G^\pi(t, x) := \int_t^T \gamma^{s-t} r(s, X_s^\pi)\,\mathrm{d}s + \gamma^{T-t} f(X_T^\pi) \quad \text{with} \quad X_t^\pi = x, \tag{2.4}$$

where $f \equiv 0$ when $\mathsf{T} = [0, \infty)$. We distinguish returns earned by $(h, a)$-persistent modifications of policies. We call these *$h$-dependent action-conditioned returns* and denote them by $Z_h^\pi(t, x, a)$. Given $\pi$, they are defined by

$$Z_h^\pi(t, x, a) := \int_t^T \gamma^{s-t} r(s, X_s^{\pi|_{h,a,t}})\,\mathrm{d}s + \gamma^{T-t} f(X_T^{\pi|_{h,a,t}}) \quad \text{with} \quad X_t^{\pi|_{h,a,t}} = x. \tag{2.5}$$

Value-based approaches in RL estimate either the *value function* $V^\pi(t,x) := \mathbf{E}[G^\pi(t,x)]$ or the *$h$-dependent action-value function* $Q_h^\pi(t,x,a) := \mathbf{E}[Z_h^\pi(t,x,a)]$.[5] As distributional approaches in RL estimate the laws of returns, following [41], we define

$$\eta^\pi(t,x) := \text{law}(G^\pi(t,x)) \quad \text{and} \quad \zeta_h^\pi(t,x,a) := \text{law}(Z_h^\pi(t,x,a)).$$

It is important to note that only the laws of random returns (and not their representations as random variables) are observable and modeled in practice.

## 2.2 $Q$-Learning in Continuous Time

The failure of action-value-based RL in continuous-time stems from the collapse of action values at a given state to the value of that state. Precisely, Tallec et al. and Jia and Zhou established that

$$Q_h^\pi(t,x,a) - V^\pi(t,x) = (H^\pi(t,x,a) + (\log\gamma)V^\pi(t,x))h + o(h), \tag{2.6}$$

where $H^\pi \in \mathbb{R}$ is independent of $h$ (see [34] and [16] respectively).[6] In a discrete action space, given a state $x$ and time $t$, a concise way to capture the asymptotic information of (2.6) is by considering the *action gap* [11, 5] of the associated $h$-dependent action values

$$\mathsf{gap}(Q_h^\pi, t, x) := \min_{a_1 \neq a_2} |Q_h^\pi(t,x,a_1) - Q_h^\pi(t,x,a_2)|.$$

Anticipating (2.6), which implies that $\mathsf{gap}(Q_h^\pi, t, x) = O(h)$, Baird proposed AU wherein he estimated the *rescaled advantage function* $A_h^\pi$ in place of $Q_h^\pi$:

$$A_h^\pi(t,x,a) := \frac{Q_h^\pi(t,x,a) - V^\pi(t,x)}{h} \quad \forall (t,x,a) \in \mathsf{T} \times \mathsf{X} \times \mathsf{A}. \tag{2.7}$$

Note that $\mathsf{gap}(A_h^\pi, t, x) = O(1)$. Tallec et al. [34] and Jia and Zhou [16], following Baird, also estimated $A_h^\pi$ to ameliorate $Q$-learning in continuous time.

## 3 The Distributional Action Gap

In this section, we define a distributional notion of action gap; we prove that $h$-dependent action-conditioned return distributions collapse to their underlying policy's return distribution as $h$ vanishes; and we quantify the rate of collapse of these return distributions.

**Definition 3.1.** *Consider an MDP with discrete action space and let $\mu : \mathsf{T} \times \mathsf{X} \times \mathsf{A} \to (\mathscr{P}(\mathbb{R}), d)$ for a metric $d$. The $d$ action gap of $\mu$ at a state $x$ and time $t$ is given by*

$$\mathsf{distgap}_d(\mu, t, x) := \min_{a_1 \neq a_2} d(\mu(t,x,a_1), \mu(t,x,a_2)).$$

While $\mathbb{R}$ has a canonical metric, induced by $|\cdot|$, the space $\mathscr{P}(\mathbb{R})$ does not. So a choice must be made, and some metrics are unsuitable. For example, in deterministic MDPs with deterministic policies, return distributions are identified by expected returns: $\zeta_h^\pi(t,x,a)$ is the delta at $Q_h^\pi(t,x,a)$, for all $(t,x,a) \in \mathsf{T} \times \mathsf{X} \times \mathsf{A}$. Thus, $\mathsf{distgap}_d(\zeta_h^\pi, t, x)$ should vanish as $h$ decreases to zero if $\mathsf{gap}(Q_h^\pi, t, x)$ vanishes as $h$ decreases to zero. With the total variation metric $d = \mathrm{TV}$, for instance, this is not the case, making TV unsuitable. Indeed, suppose we have a deterministic MDP with $\mathsf{A} = \{a_1, a_2\}$ and such that $\zeta_h^\pi(t,x,a_1) = \delta_h$ and $\zeta_h^\pi(t,x,a_2) = \delta_0$, for some state $x$ and time $t$ (see, e.g., [34]). Then $\mathsf{distgap}_{\mathrm{TV}}(\zeta_h^\pi, t, x) = 1$, for all $h > 0$, yet $\mathsf{gap}(Q_h^\pi, t, x) = h$.

The $W_p$ *distances* from the theory of Optimal Transportation (see [36]), however, are suitable. They are defined via *couplings* of distributions.

**Definition 3.2.** *Let $\mu, \nu \in \mathscr{P}(\mathbb{R})$. A $\kappa \in \mathscr{P}(\mathbb{R}^2)$ is a* coupling *of $\mu$ and $\nu$ if its first and second marginals are $\mu$ and $\nu$ respectively. We denote the set of these couplings by $\mathscr{C}(\mu, \nu)$.*

**Definition 3.3.** *Let $\mu, \nu \in \mathscr{P}(\mathbb{R})$[7] and $p \in [1, \infty)$. The $W_p$ distance between $\mu$ and $\nu$ is*

$$W_p(\mu, \nu) := \inf_{\kappa \in \mathscr{C}(\mu,\nu)} \left( \int_{\mathbb{R}^2} |z - w|^p \, \kappa(\mathrm{d}z\mathrm{d}w) \right)^{1/p}. \tag{3.1}$$

---

[5] In [38] and [16], the authors established $V^\pi$ and $Q_h^\pi$ as continuous-time RL analogues of the classic value and action-value functions. We note, however, that this was done without explicitly defining $G^\pi$ and $Z_h^\pi$.

[6] The function $H^\pi$ is the *Hamiltonian* of the classic stochastic optimal control problem defined by our MDP.

[7] Here we assume that $\mu$ and $\nu$ have finite absolute $p$th moments.

Any coupling attaining the infimum in (3.1) is called a $W_p$-*optimal coupling*. Henceforth, we write $\mathsf{distgap}_p$ when considering $W_p$ action gaps. If $\mu$ and $\nu$ are deltas at $Q_h^\pi(t,x,a_1)$ and $Q_h^\pi(t,x,a_2)$ respectively, then the right-hand side of (3.1) is equal to $|Q_h^\pi(t,x,a_1) - Q_h^\pi(t,x,a_2)|$. Hence, in deterministic MDPs with deterministic policies, $W_p$ action gaps of $\zeta_h^\pi$ are identical to action gaps of $Q_h^\pi$, making the $W_p$ distances suitable in the above sense. In non-deterministic MDPs, the relationship between $\mathsf{distgap}_p(\zeta_h^\pi, t, x)$ and $\mathsf{gap}(Q_h^\pi, t, x)$ is opaque.

The following results study the $W_p$ action gap of $\zeta_h^\pi$ as a function of $h$, lending some color to the relationship between $\mathsf{distgap}_p(\zeta_h^\pi, t, x)$ and $\mathsf{gap}(Q_h^\pi, t, x)$. These results all hold under Assumptions 2.2 and 2.3. Henceforth, we suppress mention of these assumptions; we do not restate them explicitly. First, we observe that $W_p$ action gaps of $\zeta_h^\pi$ are bounded from below by action gaps of $Q_h^\pi$.

**Proposition 3.4.** *For all* $(t,x) \in \mathsf{T} \times \mathsf{X}$*, we have that* $\mathsf{distgap}_p(\zeta_h^\pi, t, x) \geq \mathsf{gap}(Q_h^\pi, t, x)$*.*

For a proof of this statement and any other made in this work, see Appendix B. Our next result establishes that $W_p$ action gaps of $\zeta_h^\pi$, like action gaps of $Q_h^\pi$, vanishes for a large class of MDPs.

**Theorem 3.5.** *If $r$ and $f$ are bounded, then* $\lim_{h \downarrow 0} W_p(\zeta_h^\pi(t,x,a), \eta^\pi(t,x)) = 0$*, for all* $(t,x,a) \in \mathsf{T} \times \mathsf{X} \times \mathsf{A}$*; hence,* $\lim_{h \downarrow 0} \mathsf{distgap}_p(\zeta_h^\pi, t, x) = 0$*.*

While Theorem 3.5 shows that the $W_p$ distance between $\zeta_h^\pi(t,x,a)$ and $\eta^\pi(t,x)$ (and the $W_p$ action gap of $\zeta_h^\pi$ at $(t,x) \in \mathsf{T} \times \mathsf{X}$) does indeed vanish as $h$ decreases, it does not identify the rate at which it does so. Our next two theorems establish this rate.

**Theorem 3.6.** *MDPs and policies exist in and under which, for all* $(t,x,a) \in \mathsf{T} \times \mathsf{X} \times \mathsf{A}$*, we have that* $W_p(\zeta_h^\pi(t,x,a), \eta^\pi(t,x)) \gtrsim h^{1/2}$ *and* $\mathsf{distgap}_p(\zeta_h^\pi, t, x) \gtrsim h^{1/2}$*.*

Finally, we prove that for a large class of MDPs (different from but overlapping with the class of MDPs captured in Theorem 3.5), the lower bound found in Theorem 3.6 is an upper bound.

**Theorem 3.7.** *If $r$ is Lipschitz in state, uniformly in time, $f$ is Lipschitz, and $T < \infty$, then* $W_p(\zeta_h^\pi(t,x,a), \eta^\pi(t,x)) \lesssim h^{1/2}$*, for all* $(t,x,a) \in \mathsf{T} \times \mathsf{X} \times \mathsf{A}$*; hence,* $\mathsf{distgap}_p(\zeta_h^\pi, t, x) \lesssim h^{1/2}$*.*

Theorems 3.6 and 3.7 demonstrate that the $W_p$ distance between $\zeta_h^\pi(t,x,a)$ and $\eta^\pi(t,x)$ and the distance between $Q_h^\pi(t,x,a)$ and $V^\pi(t,x)$ are of different orders in terms of $h$. Thus, we see that $\mathsf{distgap}_p(\zeta_h^\pi, t, x)$ and $\mathsf{gap}(Q_h^\pi, t, x)$ in stochastic MDPs are fundamentally different.

# 4 Distributional Superiority

In this section, we introduce a probabilistic generalization of the advantage. We define this random variable—which we call the *superiority* and denote by $S_h^\pi$—via a pair of axioms.

A natural construction of the superiority at $(t,x,a)$ is given by $Z_h^\pi(t,x,a) - G^\pi(t,x)$. The law of this difference, however, depends on the joint law of $(Z_h^\pi(t,x,a), G^\pi(t,x))$, which is unobservable in practice and ill-defined (cf. Section 2.1.2). Yet, the set of all possible laws of this difference is easily characterized; it is the set of *coupled difference representations of* $\zeta_h^\pi(t,x,a)$ *and* $\eta^\pi(t,x)$.

**Definition 4.1.** *Let $\mu, \nu \in \mathscr{P}(\mathbb{R})$. A coupled difference representation (CDR) $\psi \in \mathscr{P}(\mathbb{R})$ of $\mu$ and $\nu$ takes the form $\psi = \Delta_\# \kappa$ where $\kappa \in \mathscr{C}(\mu, \nu)$ and $\Delta : \mathbb{R}^2 \to \mathbb{R}$ is given by $\Delta(z,w) := z - w$. The set of all coupled difference representations of $\mu$ and $\nu$ will be denoted by $\mathscr{D}(\mu, \nu)$.*

Our first axiom places the superiority's law in this set, $\mathscr{D}(\zeta_h^\pi(t,x,a), \eta^\pi(t,x))$.

**Axiom 1.** *The law of $S_h^\pi(t,x,a)$ is a coupled difference representation of $\zeta_h^\pi(t,x,a)$ and $\eta^\pi(t,x)$.*

Our second axiom encodes a type of consistency for deterministic policy behavior.

**Axiom 2.** *$S_h^\pi(t,x,a)$ is deterministic whenever $\pi$ is $(h,a)$-persistent at time $t$.*

To see how Axiom 2 encodes a notion of deterministic consistency, first consider its discrete-time analogue: *the superiority at $(x,a)$ for a policy $\pi$ is deterministic if $\pi$ at $x$ deterministically chooses $a$.* In this situation, our $a$-following agent makes the same choices as a $\pi$-following agent—both take action $a$ in state $x$ initially and then follow $\pi$ thereafter—, and we posit that the superiority should not be random. The continuous-time analogue of the situation just described occurs precisely when a policy $\pi$ is $(h,a)$-persistent at starting time $t$. Given a starting time $t$ and state $x$, an agent that chooses action $a$ between $t$ and $t+h$ and then follows $\pi$ is following the $(h,a)$-persistent modification of $\pi$ at

time $t$. By definition, they make the same choices as a $\pi$-following agent when $\pi$ is $(h,a)$-persistent starting at time $t$. Axiom 2 stipulates that, in this case, $S_h^\pi(t,x,a)$ should be deterministic.

By construction, if $\psi_h^\pi(t,x,a) \in \mathscr{D}(\zeta_h^\pi(t,x,a), \eta^\pi(t,x))$, then its mean is $Q_h^\pi(t,x,a) - V^\pi(t,x)$ (see Appendix B.2 for a proof of this claim). Axiom 2 then says that any determining coupling $\kappa_h^\pi(t,x,a)$ when $\pi$ is $(h,a)$-persistent at time $t$ must be such that $\Delta_\#\kappa_h^\pi(t,x,a) = \delta_0$.[8] In particular, Axiom 2 nontrivially restricts $\mathscr{D}(\zeta_h^\pi(t,x,a), \eta^\pi(t,x))$.

**Example 4.2.** *Let $\iota_h^\pi(t,x,a) := \Delta_\#\kappa_h^\pi(t,x,a)$ for $\kappa_h^\pi(t,x,a) = \zeta_h^\pi(t,x,a) \otimes \eta^\pi(t,x)$. If $\pi$ is $(h,a)$-persistent at time $t$, then $\mathbf{Var}(\iota_h^\pi(t,x,a)) = 2\mathbf{Var}(\eta^\pi(t,x))$. This variance is 0 only when $\pi$'s return is deterministic. Hence, $\iota_h^\pi(t,x,a)$ may be nontrivial even when conditioning on $a$ reflects the policy's behavior exactly. We posit, via Axiom 2, that this should be prohibited.*

In fact, Axiom 2 determines a single coupling, if we want a consistent choice across all time-state-action triplets and all MDPs.

**Theorem 4.3.** *Let $\kappa \in \mathscr{C}(\mu, \mu)$ for some $\mu \in \mathscr{P}(\mathbb{R})$. The push-forward of $\kappa$ by $\Delta$ is the delta at zero, $\Delta_\#\kappa = \delta_0$, if and only if $\kappa$ is a $W_p$-optimal coupling, for some $p \in [1, \infty)$. Moreover, there is only one such coupling. It is given by $\kappa_\mu := (\mathrm{id}, \mathrm{id})_\#\mu$ or, equivalently, $\kappa_\mu := (F_\mu^{-1}, F_\mu^{-1})_\#\mathcal{U}(0,1)$. Here $\mathcal{U}(0,1)$ is the uniform distribution on $[0,1]$.*

The second definition of $\kappa_\mu$ corresponds, more generally, to the $W_p$-optimal coupling, for all $p \geq 1$, of $\mu$ and $\nu$ given by $\kappa_{\mu,\nu} := (F_\mu^{-1}, F_\nu^{-1})_\#\mathcal{U}(0,1)$. As $\Delta_\#\kappa_{\mu,\nu}$'s quantile function is $F_\mu^{-1} - F_\nu^{-1}$, we have in hand everything we need to define the superiority distribution (via its quantile function).

**Definition 4.4.** *The* superiority distribution *$\psi_h^\pi$ at $(t,x,a) \in \mathsf{T} \times \mathsf{X} \times \mathsf{A}$ is*

$$\psi_h^\pi(t,x,a) := (F_{\zeta_h^\pi(t,x,a)}^{-1} - F_{\eta^\pi(t,x)}^{-1})_\#\mathcal{U}(0,1).$$

As $\psi_h^\pi(t,x,a)$ has the smallest possible central absolute $p$th moments among all CDRs of $\zeta_h^\pi(t,x,a)$ and $\eta^\pi(t,x)$, heuristically, it captures more of the individual features of both return distributions than other such CDRs (like $\iota_h^\pi(t,x,a)$ in Example 4.2). We illustrate this by example in Figure 4.1.

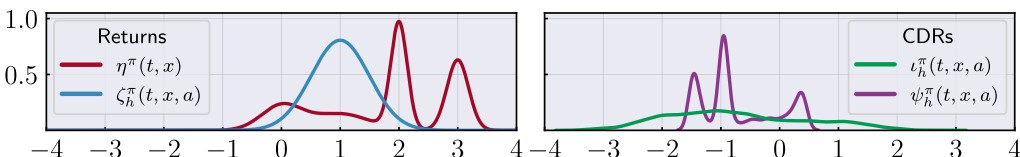

Figure 4.1: PDFs of Return Distributions and Two Candidate CDRs.

## 4.1 The Rescaled Superiority Distribution

From Section 3, we know that the $W_p$ distance between $\zeta_h^\pi(t,x,a)$ and $\eta^\pi(t,x)$, for every $p \geq 1$, vanishes as $h$ vanishes. Moreover, we know the rate at which this distance disappears. As a result, by construction, the central absolute $p$th moments of $\psi_h^\pi(t,x,a)$ collapse as $h$ collapses, and, for a large class of MDPs, we understand the rate at which these moments collapse. More generally and precisely, if we consider the *$q$-rescaled superiority distribution* defined by

$$\psi_{h;q}^\pi := (h^{-q}\,\mathrm{id})_\#\psi_h^\pi,$$

we see that Theorems 3.6 and 3.7 translate to the follow statements on $W_p$ action gaps of $\psi_{h;q}^\pi$:

**Theorem 4.5.** *MDPs and policies exist satisfying Assumptions 2.2 and 2.3 in and under which, for all $(t,x) \in \mathsf{T} \times \mathsf{X}$, we have that $\mathsf{distgap}_p(\psi_{h;q}^\pi, t, x) \gtrsim h^{1/2-q}$.*

**Theorem 4.6.** *Under Assumptions 2.2 and 2.3, if $r$ is Lipschitz in state, uniformly in time, $f$ is Lipschitz, and $T < \infty$, then $\mathsf{distgap}_p(\psi_{h;q}^\pi, t, x) \lesssim h^{1/2-q}$, for all $(t,x) \in \mathsf{T} \times \mathsf{X}$.*

These two theorems tell us how to preserves the $W_p$ action gaps of $q$-rescaled superiority distributions (as a function of $h$). They identify $q = 1/2$. For $q < 1/2$, $W_p$ action gaps vanish as $h$ vanishes. Whereas for $q > 1/2$, $W_p$ action gaps blow up as $h$ vanishes. These behaviors are undesirable. When

---

[8] Here $\zeta_h^\pi(t,x,a) = \eta^\pi(t,x)$; so the mean of $\Delta_\#\kappa_h^\pi(t,x,a)$ is 0.

$q < 1/2$, the influence of an action on an agent's superiority becomes indistinguishable from any other action. For $q > 1/2$, ever larger sample sizes are need to obtain any statistical estimate of an agent's superiority with the same level of accuracy. These scenarios are untenable.

Another consideration regarding rescalings of $\psi_h^\pi$ is whether they upend rankings of actions determined by some given measure of utility. This would be counterproductive. In DRL, agents often use *distortion risk measures* [1] to rank actions ([10, 9, 22, 4, 17]).

**Definition 4.7.** *Given* $\beta \in \mathscr{P}([0,1])$, *the* distortion risk measure $\rho_\beta : \mathscr{P}(\mathbb{R}) \to \mathbb{R}$ *is defined by* $\rho_\beta(\mu) := \langle \beta, F_\mu^{-1} \rangle$; *on* $\mu \in \mathscr{P}(\mathbb{R})$, *its value is given by the integral of* $F_\mu^{-1}$ *with respect to* $\beta$.

A family of $\rho_\beta$ is the $\alpha$-conditional value-at-risk measures ($\alpha$-CVaR) [29], where $\beta_\alpha = \mathcal{U}(0,\alpha)$ for $\alpha \in (0,1]$; $\alpha = 1$ is the expected-value utility measure. Crucially, $\psi_{h;q}^\pi$ preserves $\rho_\beta$-valued utility.

**Theorem 4.8.** *Let* $\rho_\beta$ *be a distortion risk measure,* $q \geq 0$, *and* $h > 0$. *If* $\rho_\beta(\eta^\pi(t,x)) < \infty$, *then* $\arg\max_{a \in A} \rho_\beta(\psi_{h;q}^\pi(t,x,a)) = \arg\max_{a \in A} \rho_\beta(\zeta_h^\pi(t,x,a))$.

In turn, the $1/2$-rescaled superiority distribution is not only $W_p$ action gap preserving but matches $\zeta_h^\pi$ in its greedy choice of action as measured by a distortion risk measure.

## 4.2 Algorithmic Considerations

We now turn to building DRL algorithms based on our theory. Our algorithms leverage the quantile TD-learning framework [10] to learn $\rho_\beta$-greedy policies, for a given distortion risk measure $\rho_\beta$, just as DAU [34] leverages the $Q$-learning framework to learn greedy policies. Full pseudocode and implementation details are given in Appendix C.

At the heart of our algorithms is an equality of quantile functions, which holds by construction,

$$F_{\zeta_h^\pi(t,x,a)}^{-1} = F_{\eta^\pi(t,x)}^{-1} + h^q F_{\psi_{h;q}^\pi(t,x,a)}^{-1}. \tag{4.1}$$

Indeed, given $\eta$ and $\psi_{h;q}$, as models of $\eta^\pi$ and $\psi_{h;q}^\pi$ respectively, equation (4.1) justifies the application of quantile TD-learning to $\zeta_h$, as a model for $\zeta_h^\pi$, defined via the quantile function

$$F_{\zeta_h(t,x,a)}^{-1} := F_{\eta(t,x)}^{-1} + h^q F_{\psi_{h;q}(t,x,a)}^{-1}. \tag{4.2}$$

That said, we cannot realize quantile TD-learning without defining predictions and bootstrap targets in terms of $m$-quantile representations[9] [10, 4] of $\zeta_h$, via those of $\eta$ and $\psi_{h;q}$.

While we may freely parameterize the $m$-quantile representation of $\eta$ with a neural network (with interface) $\theta : T \times X \to \mathbb{R}^m$, we have to be careful when parameterizing the $m$-quantile representation of $\psi_{h;q}$. Given a neural network $\phi : T \times X \times A \to \mathbb{R}^m$, we set

$$F_{\psi_{h;q}(t,x,a)}^{-1} := \phi(t,x,a) - \phi(t,x,a^\star) \quad \text{with} \quad a^\star \in \arg\max_{a \in A} \rho_\beta(\phi(t,x,a)). \tag{4.3}$$

This ensures we identify a $\rho_\beta$-greedy policy; it is 0 at the $\rho_\beta$-greedy action $a^\star$ (cf. [34, Eq. 27]).

With appropriate parameterized $m$-quantile representations of $\eta$ and $\psi_{h;q}$ in hand, we derive our predictions and bootstrap targets. By (4.2), recalling $\theta$ and (4.3), we compute our predictions via

$$F_{\zeta_h(t,x,a)}^{-1} := \theta(t,x) + h^q(\phi(t,x,a) - \phi(t,x,a^\star)). \tag{4.4}$$

By Wiltzer [40], as $X_t^{\pi|h,a,t} = x$ and $X_{t+h}^{\pi|h,a,t} = X_{t+h}^a$, observe that

$$Z_h^\pi(t,x,a) =_{\text{law}} \int_0^h \gamma^s r(t+s, X_{t+s}^{\pi|h,a,t}) \, \mathrm{d}s + \gamma^h G^\pi(t+h, X_{t+h}^{\pi|h,a,t})$$

$$=_{\text{law}} hr(t,x) + \gamma^h G^\pi(t+h, X_{t+h}^a) + Y_h,$$

where $\mathbf{E}[|Y_h|^p] = o(h)$[10] for all $p \in \mathbb{N}$. So upon getting a sample state/realization $x_{t+h}$ of $X_{t+h}^a$, as in [30], we compute our bootstrap targets via

$$\mathcal{T} F_{\zeta_h(t,x,a)}^{-1} := hr(t,x) + \gamma^h \theta(t+h, x_{t+h}). \tag{4.5}$$

---

[9] An $m$-quantile representation of a given distribution in $\mathscr{P}(\mathbb{R})$ is a vector in $\mathbb{R}^m$ whose $i$th component encodes the $\frac{i-1/2}{m}$-quantile of said distribution.

[10] This holds if $r$ is continuous, a standard assumption in continuous-time control (see, e.g., [25, 34, 16]).

In summary, the predictions (4.4) and the bootstrap targets (4.5) together characterize a family of QR-DQN-based algorithms called DSUP($q$), whose core update is outlined in Algorithm 1.

---

**Algorithm 1** DSUP($q$) Update

---

**Require:** $q$ (rescale factor), $\alpha$ (step size), $\mathcal{D}$ (replay buffer), $\rho_\beta$ (distortion risk measure)
**Require:** $\theta : \mathsf{T} \times \mathsf{X} \to \mathbb{R}^m$ ($m$-quantile approximator of $F_\eta^{-1}$), $\phi : \mathsf{T} \times \mathsf{X} \times \mathsf{A} \to \mathbb{R}^m$ (see (4.3))

 Sample $(t, x_t, a_t, r_t, x_{t+h}, \mathsf{done}_{t+h})$ from $\mathcal{D}$
 $a^\star \leftarrow \arg\max_a \rho_\beta(\frac{1}{m} \sum_{n=1}^m \delta_{\phi(t,x_t,a)_n})$       ▷ Risk-sensitive greedy action
 $F_\zeta^{-1}(\phi, \theta) \leftarrow \theta(t, x_t) + h^q(\phi(t, x_t, a_t) - \phi(t, x_t, a^\star))$       ▷ Prediction (4.4)
 $\mathcal{T}F_\zeta^{-1} \leftarrow hr_t + \gamma^h(1 - \mathsf{done}_{t+h})\theta(t + h, x_{t+h}) + \gamma^h \mathsf{done}_{t+h} f(x_{t+h})$       ▷ Target (4.5)
 $\ell(\phi, \theta) \leftarrow \mathsf{QuantileHuber}(F_\zeta^{-1}(\phi, \theta), \mathcal{T}F_\zeta^{-1})$       ▷ See [10, Eq. (10)]
 $\phi \leftarrow \phi - \alpha \nabla_\phi \ell(\phi, \theta)$ and $\theta \leftarrow \theta - \alpha \nabla_\theta \ell(\phi, \theta)$       ▷ Gradient updates

---

One theoretical drawback of DSUP($q$) for mean-return control is that the mean of the $q$-rescaled superiority distribution is $O(1)$ only when $q = 1$, by (2.6) and (2.7). Thus, we propose modeling $A_h^\pi$ simultaneously. This yields a novel form of a *two-timescale* approach to value-based RL (see, e.g., [8]). In particular, we estimate $\vartheta_{h;q}^\pi$ defined by

$$F_{\vartheta_{h;q}^\pi(t,x,a)}^{-1} := F_{\psi_{h;q}^\pi(t,x,a)}^{-1} + (1 - h^{1-q})A_h^\pi(t, x, a).$$

We call $\vartheta_{h;q}^\pi$ the *advantage-shifted $q$-rescaled superiority*. Note that its mean is $A_h^\pi$, which is $O(1)$. To realize this, we approximate $A_h^\pi$ using DAU and employ parameter sharing between the approximators of $A_h^\pi$ and $\psi_{h;q}^\pi$. We call this family of algorithms DAU+DSUP($q$). We note that $A_h^\pi$ is used only for increasing action gaps; it does not change the training loss for $\eta$ and $\psi_{h;q}$.

## 5 Simulations

The empirical work herein is two-fold in nature: illustrative and comparative. First, we simulate an example that illustrates Theorems 3.6/4.5 and Theorem 3.7/4.6 and their consequences. Second, in an option-trading environment, we compare the performance of $\psi_h^\pi$-based agent(s) against QR-DQN [10] and DAU [34] in the risk-neutral setting and against QR-DQN in a risk-sensitive setting.

### 5.1 The Rescaled Superiority Distribution Revisited

Consider an MDP with time horizon $10$, a two element action space, $0$ and $1$—when action $1$ is executed, the system follows 1-dimensional Brownian dynamics with a constant drift of $10$, and otherwise, the state is fixed—, a reward that equals the agent's signed distance to $0$, and a trivial terminal reward. We estimate four distributions at $(t, x, a) = (0, 0, 1)$ for the policy that always selects $0$. Figure 5.1 shows these estimated distributions for a sample of frequencies (kHz), $\omega = 1/h$.

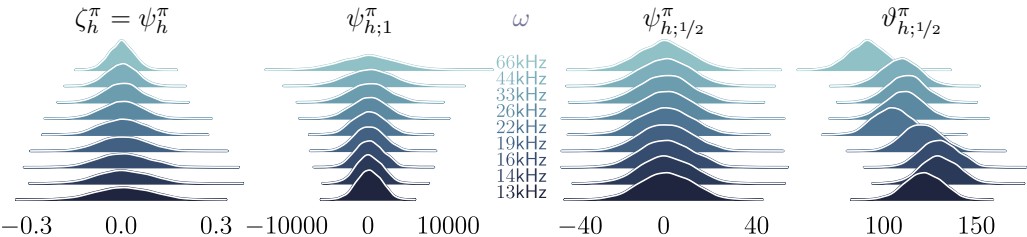

Figure 5.1: Monte-Carlo estimates of $\psi_{h;q}^\pi$, for $q = 0, 1, 1/2$, and $\vartheta_{h;1/2}^\pi$ as a function of $\omega = 1/h$.

First (from the left), we see that $\psi_h^\pi$ collapses to $\delta_0$, as $h$ tends to $0$. Thus, accurate action ranking distributional or otherwise becomes impossible in the vanishing $h$ limit. Second, we see that rescaling by $h$ produces distributions with $O(1)$ mean but infinite non-mean statistics in the vanishing $h$ limit. Here the $O(1)$ means are imperceptible in face of the large variances. So while this rescaling permits ranking actions by action values, it does so at the expense of producing high-variance distributions.

Third, we see that rescaling by $h^{1/2}$ yields distributions with $O(1)$ non-mean statistics but vanishingly small means, $O(h^{1/2})$. Hence, this rescaling permits ranking actions by non-mean statistics, even if action values again becomes indistinguishable in the vanishing $h$ limit. That said, the vanishing rate of the means here is slower than when no rescaling is considered, $O(h)$. Fourth, we see that rescaling by $h^{1/2}$ and then shifting it by $(1 - h^{1/2})A_h^\pi$ produces distributions with $O(1)$ mean and non-mean statistics. In turn, this two-timescale approach permits ranking actions by either action values or non-mean statistics (but not both by Theorem 4.8). However, the mean estimates here are inaccurate and imprecise—rather than uniformly being 100, they oscillate substantially.

In risk-neutral control, we are left with a number of questions. What effect do the high variance distributions in DAU/DSUP(1) have on performance? What effect do the $O(h^{1/2})$ means have on the performance of DSUP($1/2$)? What effect does the instability of the mean estimates in DAU+DSUP($1/2$) have on performance? In Section 5.2, we begin to answer these questions and others by testing our superiority-based algorithms against appropriate benchmarks in an option-trading environment.

## 5.2 High-Frequency Option Trading

The option-trading environment in which we run our comparative experiments is a commonly used benchmark (see, e.g., [22, 17]). We use an Euler–Maruyama discretization scheme [23] at high resolution to simulate high-frequency trading. Returns are averaged over 10 seeds and 10 different dynamics models (corresponding to data from different stocks). Additionally, following [22], we use disjoint datasets to estimate the dynamics parameters for simulation during training and evaluation.[11]

First, we consider the risk-neutral setting. Here we compare QR-DQN, DAU, and three algorithms based on the $q$-rescaled superiority distribution with $q = 1, 1/2$: DSUP(1), DAU+DSUP($1/2$), and DSUP($1/2$). Figure 5.2 summarizes their performance at a sample of frequencies (Hz).

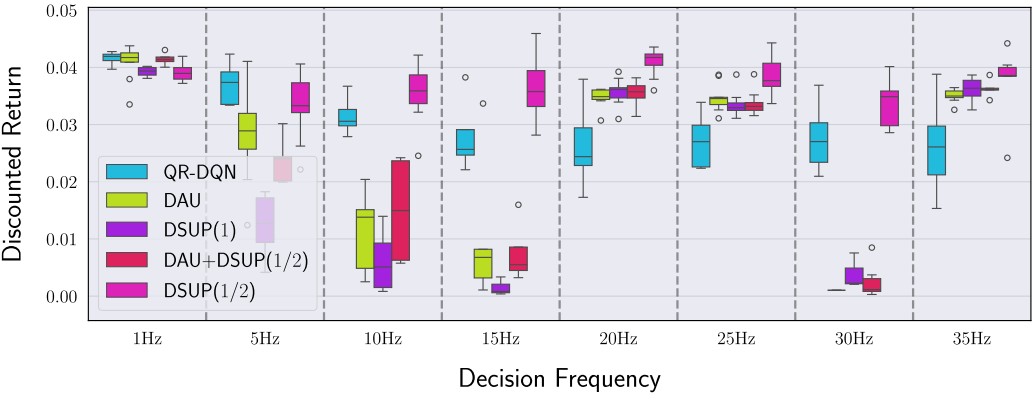

Figure 5.2: Risk-neutral algorithms on high-frequency option-trading as a function of $\omega$.

We see that DSUP($1/2$) is not only the most consistent performer, but outperforms every competitor at all but the two lowest frequencies. Even then, its performance is very close to the best performer. We also see that DAU+DSUP($1/2$)'s preservation of both action gaps and $W_p$ action gaps does not lead to the strongest performance. In particular, its performance is inconsistent and sometimes poor. We believe this is because the tested frequencies are low enough that DSUP($1/2$) maintains large enough action gaps to learn performant policies, but high enough that the variances of the distributions underlying $A_h^\pi$ cause estimation difficulty. Indeed, the three methods that estimate $A_h^\pi$ (explicitly in DAU and DAU+DSUP($1/2$) or implicitly in DSUP(1)) exhibit almost identical behavior.

Our results highlight a dichotomy in existing (ours included) methods for value-based, high-frequency, risk-neutral control. They can either maintain $O(1)$ expected return estimates or $O(1)$ return variance estimates, but not both. We observe better performance in estimating small means from $O(1)$ variance distributions than in estimating $O(1)$ means from receipricolly large variance distributions.

To qualitatively illustrate the appeal of the $1/2$-rescaled superiority, Figure 5.3 presents examples of learned action-conditioned distributions used by DSUP($1/2$) and QR-DQN agents to make decisions.

[11] Our code is available at `https://github.com/harwiltz/distributional-superiority`.

In this environment, action 1 taken in the start state terminates the episode, yielding the smallest return, 0, making this action inferior to its alternative action, 1. We see that DSUP($1/2$) infers this fact. QR-DQN, on the other hand, has difficulty distinguishing these actions. This is because the $1/2$-rescaled superiority preserves $W_p$ action gaps, while $\zeta_h^\pi$ does not.

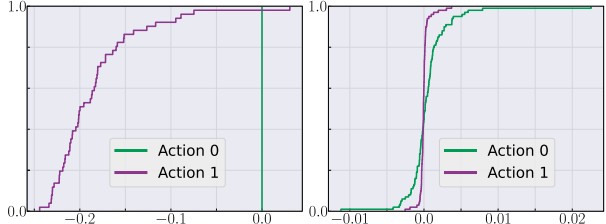

Figure 5.3: CDFs of $\psi_{h;1/2}^\pi$ from DSUP($1/2$) (left) and $\zeta_h^\pi$ from QR-DQN (right) at the start state at $\omega = 35$Hz.

Second, we consider a risk-sensitive setting. Here we compare QR-DQN and DSUP($1/2$) using $\alpha$-CVaR for greedy action selection. We do this because Theorem 4.8 does not hold with $\vartheta_{h;q}^\pi$, and preserving means is less critical in risk-sensitive control than it is in risk-neutral control. Figure 5.4 depicts our results at $\omega = 35$Hz (see Appendix D for results across a range of $\omega$).

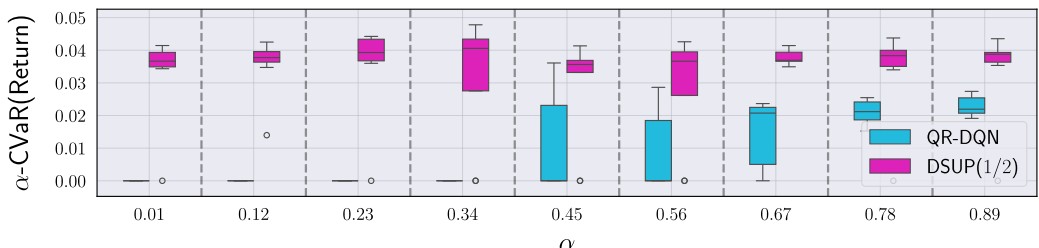

Figure 5.4: Risk-sensitive algorithms on high-frequency option-trading at $\omega = 35$Hz.

Again, we see that DSUP($1/2$) is conclusively the best performer.

## 6 Related Work

Notions of action gap and ranking have long been of interest in RL (see, e.g., [11]). Action gaps are related to sample complexity in RL—indeed, instance-dependent sample complexity rates are inversely proportional to the divergence between action-conditioned return distributions ([13, 19, 37]). Bellemare et al. [5] argue for the consideration of alternatives to the Bellman operator that explicitly devalue suboptimal actions, and they show that Baird's AL [2] operator falls within this class of operators. On the other hand, Schaul et al. [31] implicitly question Bellemare et al.'s position. They demonstrate that stochastic gradient updates in deep value-based RL algorithms induce frequent changes in relative action values, which in turn is a mechanism for exploration.

The advantage function is commonplace in RL (see, e.g., [39, 32, 27, 35, 24]). In [24], Mésnard et al. employ a distributional critic that is closely related to our (unscaled) distributional superiority. Their choice of critic stems from a desire to minimize variance. We note that the distributional superiority is *a posteriori* characterized as a minimal variance coupled difference representation of action-conditioned return distributions and policy-induced return distributions.

Lastly, DRL in continuous-time MDPs is in its infancy. There are only three works to mention. Wiltzer et al. [40, 41] give a characterization of return distributions for policy evaluation, and Halperin [14] studies algorithms for control. That said, neither work considers distributional notions of action gaps or advantages. Moreover, Halperin does not consider any of the challenges of estimating the influence of actions in high decision frequency settings.

## 7 Conclusion

We establish that DRL agents are sensitive to decision frequency through analysis and simulation. In experiments, DSUP($1/2$) learns well-performing policies across a range of high decision frequencies, unlike prior approaches. DSUP(1) and DAU+DSUP($1/2$) are less robust. Given our analysis, the performance of DSUP(1) is expected. Building an alternate algorithm to DAU+DSUP($1/2$) that is both tailored to risk-neutral control and robust to $h$ is an important avenue for future work.

## Acknowledgments and Disclosure of Funding

The authors are very grateful to Yunhao Tang for fruitful correspondence about distributional analogues to the advantage. Additionally, we thank Mark Rowland, Jesse Farebrother, Tyler Kastner, Pierluca D'Oro, Nate Rahn, and Arnav Jain for helpful discussions. HW was supported by the Fonds de Recherche du Québec and the National Sciences and Engineering Research Council of Canada (NSERC). MGB was supported by the Canada CIFAR AI Chair program and NSERC. This work was supported in part by DARPA HR0011-23-9-0050 to PS. YJ was supported by in part by NSF Grant 2243869. This research was enabled in part by support provided by Calcul Québec, the Digital Research Alliance of Canada (`alliancecan.ca`), and the compute resources provided by Mila (`mila.quebec`).

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

# A    Formalism of Continuous-Time RL Controlled Markov Processes

Expected-value RL is a data-driven approach to solving the (classic) optimal control problem: find an action (control) process $(A_s)_{s \geq t}$ and an associated state process (then determined by the environment) $(X_s)_{s \geq t}$ with $X_t = x$, for a given $t \geq 0$, that maximize the expected return earned by following the state-action process $(X_s, A_s)_{s \geq t}$. In particular, RL agents search the space of state-action processes via policies $\pi : \mathsf{T} \times \mathsf{X} \to \mathscr{P}(\mathsf{A})$. Policies prescribe the conditional probabilities of the laws of state-action processes. Indeed, $(X_s, A_s)_{s \geq t}$ is the state-action process of an agent following $\pi$ if and only if, for each $s \geq t$, the set $\{\pi(\cdot \,|\, s, x)\}_{x \in \mathsf{X}}$ is the set of conditional probabilities of $\mathrm{law}((X_s, A_s))$ with respect to $\mathrm{law}(X_s)$.

Continuous-time RL is a data-driven approach to stochastic optimal control. Whence, environmental dynamics are assumed to arise from an action-parameterized family of SDEs determined by a drift $b : \mathsf{T} \times \mathsf{X} \times \mathsf{A} \to \mathbb{R}^n$ and diffusion $\sigma : \mathsf{T} \times \mathsf{X} \times \mathsf{A} \to \mathbb{R}^{n \times n}$.[12] Thus, the goal of expected-value RL (and stochastic optimal control) is to find an expected-return maximizing state-action process among state-action processes $(X_s, A_s)_{s \geq t}$ that satisfy

$$\mathrm{d}X_s = b(s, X_s, A_s)\,\mathrm{d}s + \sigma(s, X_s, A_s)\,\mathrm{d}B_s \quad \text{with} \quad X_t = x. \tag{A.1}$$

We note that the MDPs defined in Section 2 have equivalent formulations in terms of transition kernel, exactly as they formulated in discrete-time RL. We refer the reader to [33] for an in-depth discussion regarding this fact.

## A.1    Justification of Random Returns

Given the above formalism, the "true" distribution of returns of an agent following a policy $\pi$ is the law of

$$\int_t^T \gamma^{s-t} r(s, X_s)\,\mathrm{d}s + \gamma^{T-t} f(X_T)$$

where $(X_s, A_s)_{s \geq t}$ is a state-action process associated to $\pi$ and solves (A.1). That said, by the definition of $\pi$,

$$b^\pi(s, X_s) = \mathbf{E}[b(s, X_s, A_s) \,|\, X_s] \quad \text{and} \quad \sigma^\pi(\sigma^\pi)^\top(s, X_s) = \mathbf{E}[\sigma\sigma^\top(s, X_s, A_s) \,|\, X_s],$$

where $b^\pi$ and $\sigma^\pi$ are exactly the coefficients defined in (2.3). Hence, by [7], provided that $b^\pi$ and $\sigma^\pi$ are regular enough to guarantee that (2.2) is well-posed in law, the processes $X^\pi = (X_s^\pi)_{s \geq t}$ and $X = (X_s)_{s \geq t}$ are equal in law.[13] Here $(X_s^\pi)_{s \geq t}$ satisfies (2.2) with $X_t^\pi = x$. Consequently,

$$G^\pi(t, x) =_{\mathrm{law}} \int_t^T \gamma^{s-t} r(s, X_s)\,\mathrm{d}s + \gamma^{T-t} f(X_T).$$

This, formally, justifies $G^\pi$ and $Z_h^\pi$, as defined in (2.4) and (2.5).

## A.2    On Assumption 2.3

Here we provide some conditions under which Assumption 2.3 is established. These conditions are presented as additional assumptions. Assumptions A.1 and A.2 are common in stochastic control theory (see, e.g., [12]) and SDE theory in general (see, e.g., [26]). Assumption A.3 is ubiquitous in the continuous-time RL literature [15, 16, 42]. Notably, these conditions together guarantee the existence of transition probabilities for policy-induced state processes arising from (2.2).

**Assumption A.1.** *The coefficients $b$ and $\sigma$ are uniformly bounded: a finite, positive constant $C_{A.1}$ exists such that*

$$\sup_{t,x,a} |b(t,x,a)| + \sup_{t,x,a} |\sigma(t,x,a)| \leq C_{A.1}.$$

**Assumption A.2.** *The matrix $\sigma\sigma^\top$ is uniformly elliptic: a positive, finite constant $\lambda_{A.2}$ exists such that*

$$\inf_{t,x,a} \inf_{|v|=1} v^\top \sigma\sigma^\top(t,x,a)v \geq \lambda_{A.2}^2 I.$$

---

[12] Without loss of generality, as the the laws of our stochastic processes are the objects of interest, we may assume that our diffusion matrix is square. Indeed, for any SDE with $n \times m$-dimensional diffusion $\sigma$, the matrix $(\sigma\sigma^\top)^{1/2}$, which is uniquely defined by $\sigma\sigma^\top$, and not $\sigma$ is the diffusion matrix that determines the law of the solution to that SDE.

[13] The assumptions under which we work guarantee that $X^\pi =_{\mathrm{law}} X$.

A consequence of Assumption A.2 is

$$\inf_{t,x} \inf_{|v|=1} v^\top \sigma^\pi(t,x)v \geq \lambda_{A.2} I. \tag{A.2}$$

In other words, $\sigma^\pi$ is also uniformly elliptic.

**Assumption A.3.** *A finite, positive constant $C_{A.3}$ exists for which*

$$\sup_t \mathrm{TV}(\pi(\cdot\,|\,t,x), \pi(\cdot\,|\,t,y)) \leq C_{A.3}|x-y| \quad \forall x,y \in \mathsf{X},$$

*where* $\mathrm{TV}$ *is the total variation metric on* $\mathscr{P}(\mathsf{A})$.

Observe if $\pi$ satisfies Assumption A.3, then $\pi|_{h,a,t}$ also satisfies Assumption A.3. Indeed,

$$\mathrm{TV}(\pi|_{h,a,t}(\cdot\,|\,s,x), \pi|_{h,a,t}(\cdot\,|\,s,y)) \leq \mathrm{TV}(\pi(\cdot\,|\,s,x), \pi(\cdot\,|\,s,y))$$

since

$$\sup_{s\in[t,t+h]} \mathrm{TV}(\pi|_{h,a,t}(\cdot\,|\,s,x), \pi|_{h,a,t}(\cdot\,|\,s,y)) = 0$$

and $\pi|_{h,a,t}(\cdot\,|\,s,x) = \pi(\cdot\,|\,s,x)$ for all $s \in \mathsf{T} \setminus [t,t+h)$.

**Proposition A.4.** *If Assumptions 2.2, A.1, and A.2 hold and $\pi$ satisfies Assumption A.3, then Assumption 2.3 holds.*

*Proof.* Observe that

$$\begin{aligned}
|b^\pi(t,x) - b^\pi(t,y)| &\leq \left| \int (b(t,x,a) - b(t,y,a))\,\pi(\mathrm{d}a\,|\,t,x) \right| \\
&\quad + \left| \int b(t,y,a)\,\pi(\mathrm{d}a\,|\,t,x) - \int b(t,y,a)\,\pi(\mathrm{d}a\,|\,t,y) \right| \\
&\leq (C_{2.2} + 2C_{A.1}C_{A.3})|x-y|,
\end{aligned}$$

by Assumptions 2.2, A.1, and A.3. Here we have also used Kantorovich duality to computed TV and invoke Assumption A.3.

Let $\lambda$ be an eigenvalue of $\sigma^\pi(t,x) - \sigma^\pi(t,y)$ with with unit eigenvector $v$. Observe that

$$\begin{aligned}
v^\top(\sigma^\pi(t,x)^2 - \sigma^\pi(t,y)^2)v &= v^\top(\sigma^\pi(t,x) - \sigma^\pi(t,y))\sigma^\pi(t,x) + \sigma^\pi(t,y)(\sigma^\pi(t,x) - \sigma^\pi(t,y))v \\
&= \lambda v^\top(\sigma^\pi(t,x) + \sigma^\pi(t,x))v
\end{aligned}$$

Hence, by (A.2),

$$|\lambda| = \frac{|v^\top(\sigma^\pi(t,x)^2 - \sigma^\pi(t,y)^2)v|}{v^\top(\sigma^\pi(t,x) + \sigma^\pi(t,x))v} \leq \frac{1}{2\lambda_{A.2}}|v^\top(\sigma^\pi(t,x)^2 - \sigma^\pi(t,y)^2)v|.$$

In turn, by Assumptions 2.2, A.1, and A.3, as done to prove that $b^\pi$ was Lipschitz above,

$$|\lambda| \leq \frac{1}{\lambda_{A.2}}(C_{2.2} + C_{A.1}C_{A.3})|x-y|.$$

Assumption 2.3 follows, since $\lambda$ was an arbitrary eigenvalue and all norms on finite dimensional spaces are equivalent. $\qquad\square$

We conclude this section with one final fact: under Assumption 2.2, the policy-averaged coefficient (2.3) have linear growth. Indeed,

$$|b^\pi(t,x)| \leq \int |b(t,x,a)|\,\pi(\mathrm{d}a\,|\,t,x) \leq C_{2.2}(1+|x|)$$

and

$$|\sigma^\pi(t,x)|^2 \leq \int |\sigma(t,x,a)|^2\,\pi(\mathrm{d}a\,|\,t,x) \leq C_{2.2}^2(1+|x|)^2.$$

### A.3 Action-Independent Rewards

In this paper, we assume that the rewards do not depend on actions. This is a theoretical limitation of not just our work, but continuous-time DRL in general (see [41, 14]). In the following sections, we discuss the nature of this theoretical limitation. However, many MDPs have action-independent reward functions. For example, MDPs encoding goal-reaching problems, tracking problems, and commodity-trading problems all have action-independent rewards.

#### A.3.1 Continuous-Time, Expected Return

In continuous-time, expected-value RL, when the reward function $r$ depends on actions, the standard approach to analysis involves considering the averaged reward function $r^\pi : \mathsf{T} \times \mathsf{X} \to \mathbb{R}$ given by

$$r^\pi(t, x) := \int_\mathsf{A} r(t, x, a)\, \pi(\mathrm{d}a \,|\, t, x).$$

In continuous-time RL specifically, the averaged reward $r^\pi$ is justified exactly as the coefficients $b^\pi$ and $\sigma^\pi$ are justified. However, the "true" return distribution is not equal to the law of

$$\int_t^T \gamma^{s-t} r^\pi(s, X_s^\pi)\, \mathrm{d}s + \gamma^{T-t} f(X_T^\pi). \tag{A.3}$$

To see this, it suffices to consider an MDP with a single state $x$. In this case, the expression in (A.3) is deterministic. However, if $\pi$ is nondeterministic and $r$ is dependent on actions, then the "true" return distribution is nondeterministic. Hence, the law of the expression in (A.3) cannot be the "true" return distribution associated to $\pi$.

#### A.3.2 Discrete-Time, Random Return

In discrete-time RL, the distribution of returns given an action-dependent reward is analyzed through the state-action process induced by a policy. This processes is defined by extending the action-parameterized family of transition probability kernels on $\mathsf{X}$, which define the dynamics of a given MDP, to a single transition probability kernel on $\mathsf{X} \times \mathsf{A}$. In the time-homogeneous setting, for instance, with transition kernels $\{P(\mathrm{d}y \,|\, x, a)\}_{a \in \mathsf{A}}$, this amounts to constructing

$$P^\pi(\mathrm{d}y\mathrm{d}b \,|\, x, a) := \pi(\mathrm{d}b \,|\, y) \otimes P(\mathrm{d}y \,|\, x, a),$$

provided that map $y \mapsto \pi(E \,|\, y)$ is measurable for all $y \in \mathsf{X}$. In continuous-time environments, such a constructing has yet to be discovered.

We note that trying to analogously extend the action-parameterized family of transition semigroups on $\mathsf{X}$, which define the dynamics of a given time-homogeneous MDP in continuous time, to a single transition semigroup on $\mathsf{X} \times \mathsf{A}$ by defining

$$P_t^\pi(\mathrm{d}y\mathrm{d}b \,|\, x, a) := \pi(\mathrm{d}b \,|\, y) \otimes P_t(\mathrm{d}y \,|\, x, a),$$

where $P_t(\mathrm{d}y \,|\, x)$ is a transition semigroup, may fail to satisfy the Chapman–Kolmogorov identity. Indeed, suppose $\mathsf{A}$ has two elements and $\mathsf{X} = \mathbb{R}$. Let $\pi(\mathrm{d}a \,|\, x)$ be the uniform measure on $\mathsf{A}$ for all $x \in \mathsf{X}$. If $P_t(\mathrm{d}y \,|\, x, a_\delta) = \delta_{x+t}(\mathrm{d}y)$ and $P_t(\mathrm{d}y \,|\, x, a_\mathrm{g}) = (2\pi)^{-1/2} \exp(-|y - x|^2/2t)\, \mathrm{d}y$, then Chapman–Kolmogorov identity fails, for example, on any tuple $(s, t, x, a_\delta, E \times F)$ where $E \subset \mathsf{X}$ is open, $x + t + s \notin E$, and $\pi(F) \neq 0$. On one hand,

$$P_{t+s}^\pi(E \times F \,|\, x, a_\delta) = \pi(F)\delta_{x+t+s}(E) = 0.$$

On the other hand,

$$\int P_s^\pi(E \times F \,|\, y, a)\, P_t^\pi(\mathrm{d}y\mathrm{d}a \,|\, x, a_\delta) = \frac{\pi(F)}{2}\big(P_s(E \,|\, x + t, a_\mathrm{g}) + \delta_{x+t+s}(E)\big) > 0.$$

So

$$P_{t+s}^\pi(E \times F \,|\, x, a_\delta) \neq \int P_s^\pi(E \times F \,|\, y, a)\, P_t^\pi(\mathrm{d}y\mathrm{d}a \,|\, x, a_\delta).$$

At present, the question of how to generate a well-defined (even in law) state-action process in any continuous-time MDP framework given a stochastic policy is generally open. Of course, if $\pi$ is deterministic, then the state-action process is $(X_s, \pi(s, X_s))_{s \geq t}$. If $b$ and $\sigma$ are Lipschitz in state and action, uniformly in time, and $\pi$ is Lipschitz in state, uniformly in time, then (A.1) with $A_s = \pi(s, X_s)$ is well-posed.

# B Proofs

## B.1 The Distributional Action Gap

In this section, we prove the statements made in Section 3.

Before proving any of the statements made in Section 3, we recall an identity that relates the $W_p$ distance between two probability measures $\mu$ and $\nu$ and the absolute central $p$th moments of the differences of random variables distributed according to $\mu$ and $\nu$:

$$W_p(\mu, \nu)^p = \inf_{(X,Y)} \{\mathbf{E}[|X - Y|^p] : \text{law}(X) = \mu \text{ and } \text{law}(Y) = \nu\}. \tag{B.1}$$

This identity will be used a number of times, including in the proof of Section 3's first result, which we restate here for the readers convenience.

**Proposition 3.4.** *For all $(t, x) \in \mathsf{T} \times \mathsf{X}$, we have that $\text{distgap}_p(\zeta_h^\pi, t, x) \geq \text{gap}(Q_h^\pi, t, x)$.*

*Proof.* Let $(Z_1, Z_2)$ be any random vector with such that $\text{law}(Z_i) = \zeta_h^\pi(t, x, a_i)$ for $i = 1, 2$. Then, by Jensen's inequality,

$$\mathbf{E}[|Z_1 - Z_2|^p]^{1/p} \geq \mathbf{E}[|Z_1 - Z_2|] \geq |\mathbf{E}[Z_1 - Z_2]| = |Q_h^\pi(t, x, a_1) - Q_h^\pi(t, x, a_2)|.$$

Hence, since $(Z_1, Z_2)$ was arbitrary, by (B.1),

$$W_p(\zeta_h^\pi(t, x, a_1), \zeta_h^\pi(t, x, a_2)) \geq |Q_h^\pi(t, x, a_1) - Q_h^\pi(t, x, a_2)|.$$

Finally, taking the minimum over pairs of actions $(a_1, a_2)$ such that $a_1 \neq a_2$ concludes the proof. $\square$

Now we move on to the proofs of Theorems 3.5 and 3.7. We defer the proof of Theorem 3.6 until after the proof of Theorem 3.7 as the proofs of Theorems 3.5 and 3.7 are similar. For clarity's sake, we first prove a collection of lemmas.

**Lemma B.1.** *Under Assumption 2.2, let $(X_s^a)_{s \geq t}$ be the unique strong solution to (2.2) with $X_t^a = x$ $\mathbf{P}$-a.s. Then, for all $q \geq 1$ and for all $s \geq t$,*

$$\mathbf{E}[|X_s^a - x|^{2q}] \leq C_q C_{2.2}^{2q}(1 + |x|)^{2q}((s - t)^q + 1)(s - t)^q e^{C_q C_{2.2}^{2q}((s-t)^q + 1)(s-t)^q}$$

*where $C_q = 4^{2q-1}$.*

*Proof.* Let $C_q = 4^{2q-1}$. By Jensen's inequality and Itô's isometry, observe that

$$
\begin{aligned}
|X_s^a - x|^{2q} &\leq C_q(s - t)^{2q-1} \int_t^s |b(s', x, a)|^{2q} \, ds' + C_q(s - t)^{q-1} \int_t^s |\sigma(s', x, a)|^{2q} \, ds' \\
&\quad + C_q(s - t)^{2q-1} \int_t^s |b(s', X_{s'}^a, a) - b(s', x, a)|^{2q} \, ds' \\
&\quad + C_q(s - t)^{q-1} \int_t^s |\sigma(s', X_{s'}^a, a) - \sigma(s', x, a)|^{2q} \, ds' \\
&\leq C_q((s - t)^{2q} + (s - t)^q) C_{2.2}^{2q}(1 + |x|)^{2q} \\
&\quad + C_q((s - t)^{2q-1} + (s - t)^{q-1}) C_{2.2}^{2q} \int_t^s |X_{s'}^a - x|^{2q} \, ds'.
\end{aligned}
$$

Thus, the lemma follows after taking expectation and applying Gronwall's inequality. $\square$

**Lemma B.2.** *Under Assumptions 2.2 and 2.3, let $(X_s^\bullet)_{s \geq t}$ with $\bullet \in \{\pi, \pi|_{h,a,t}\}$ be the unique strong solution to (2.2) with $X_t^\bullet = x$ $\mathbf{P}$-a.s. Then, for all $s \leq t + h$,*

$$\mathbf{E}[|X_s^{\pi|h,a,t} - X_s^\pi|^{2q}] \leq C(1 + |x|)^{2q}((s - t)^q + 1)(s - t)^q e^{C((s-t)^q + 1)(s-t)^q}$$

*where $C$ is some finite positive constant depending on $q$, $C_{2.2}$, and $C_{2.3}$.*

*Proof.* Let $C_q = 8^{2q-1}$. Note that $X_{s'}^{\pi|_{h,a,t}} = X_{s'}^a$ **P**-a.s. for all $s' \leq t+h$, by the definition of $\pi|_{h,a,t}$ and the uniqueness of strong solutions to (2.1). So, by Jensen's inequality and Itô's isometry, observe that

$$|X_s^{\pi|_{h,a,t}} - X_s^\pi|^{2q} \leq C_q((s-t)^{2q-1} + (s-t)^{q-1})\bigg( \int_t^s (C_{2.2}^{2q} + C_{2.3}^{2q})|X_{s'}^a - x|^{2q} \,\mathrm{d}s'$$

$$+ \int_t^s (C_{2.2}^{2q} + C_{2.3}^{2q})(1 + |x|)^{2q} \,\mathrm{d}s'$$

$$+ \int_t^s C_{2.3}^{2q}|X_{s'}^{\pi|_{h,a,t}} - X_{s'}^\pi|^{2q} \,\mathrm{d}s' \bigg).$$

Thus, after taking expectation, we deduce that

$$\mathbf{E}[|X_s^{\pi|_{h,a,t}} - X_s^\pi|^2] \leq C_q((s-t)^{2q} + (s-t)^q)\bigg( (C_{2.2}^{2q} + C_{2.3}^{2q}) \max_{s' \in [s,t]} \mathbf{E}[|X_{s'}^a - x|^{2q}]$$

$$+ (C_{2.2}^{2q} + C_{2.3}^{2q})(1 + |x|)^{2q} \bigg)$$

$$+ C_q((s-t)^{2q-1} + (s-t)^{q-1})C_{2.3}^{2q} \int_t^s \mathbf{E}[|X_{s'}^{\pi|_{h,a,t}} - X_{s'}^\pi|^{2q}] \,\mathrm{d}s'.$$

And so, the lemma follows after applying Lemma B.1 and Gronwall's inequality. $\qquad\square$

**Lemma B.3.** *Under Assumptions 2.2 and 2.3, let $(X_s^\bullet)_{s \geq t}$ with $\bullet \in \{\pi, \pi|_{h,a,t}\}$ be the unique strong solution to (2.2) with $X_t^\bullet = x$ **P**-a.s. Then, for all $s > t+h$,*

$$\mathbf{E}[|X_s^{\pi|_{h,a,t}} - X_s^\pi|^{2q}] \leq C(1 + |x|)^{2q}(h^q + 1)h^q e^{C((s-t-h)^q+1)(s-t-h)^q}$$

*where $C$ is some finite positive constant depending on $q$, $C_{2.2}$, and $C_{2.3}$.*

*Proof.* Let $C_q = 3^{2q-1}$. Observe that

$$X_s^\bullet = X_{t+h}^\bullet + \int_{t+h}^s b^\bullet(s', X_{s'}^\bullet) \,\mathrm{d}s' + \int_{t+h}^s \sigma^\bullet(s', X_{s'}^\bullet) \,\mathrm{d}B_{s'}$$

So, as $\pi|_{h,a,t}(\cdot \mid s', y) = \pi(\cdot \mid s', y)$ for all $(s', y) \in \mathsf{T} \setminus [t, t+h) \times \mathsf{X}$, by Jensen's inequality and Itô's isometry,

$$|X_s^{\pi|_{h,a,t}} - X_s^\pi|^{2q} \leq C_q|X_{t+h}^{\pi|_{h,a,t}} - X_{t+h}^\pi|^{2q} + C_q\bigg( (s-t-h)^{2q-1}$$

$$+ (s-t-h)^{q-1}\bigg)C_{2.3}^{2q} \int_{t+h}^s |X_{s'}^{\pi|_{h,a,t}} - X_{s'}^\pi|^{2q} \,\mathrm{d}s'.$$

Thus, after taking expectation, applying Gronwall's inequality, and considering Lemma B.2, the lemma follows. $\qquad\square$

One consequence of Lemmas B.2 and B.3 is

$$\mathbf{E}[|X_s^{\pi|_{h,a,t}} - X_s^\pi|^p] \to 0 \quad \text{as} \quad h \downarrow 0,$$

for all $s \in \mathsf{T}$ and for all $p \in [1, \infty)$. And so, if $f$ is bounded, then $f(X_T^{\pi|_{h,a,t}}) - f(X_T^\pi)$ is bounded and converges to zero **P**-a.s. as $h$ converges to zero. The dominated convergence theorem implies that

$$\mathbf{E}[|(f(X_T^{\pi|_{h,a,t}}) - f(X_T^\pi))|^p] \to 0 \quad \text{as} \quad h \downarrow 0. \tag{B.2}$$

Similarly, we see that the functions $g_h : \mathsf{T} \to [0, \infty)$ defined by

$$g_h(s) := \mathbf{E}[|r(s, X_s^{\pi|_{h,a,t}}) - r(s, X_s^\pi)|^p]$$

are uniformly bounded (in $h$) and converge to zero as $h \downarrow 0$ for every $s \in \mathsf{T}$. Hence, by the dominated convergence theorem, again,

$$\int_t^T \mathbf{E}[|r(s, X_s^{\pi|_{h,a,t}}) - r(s, X_s^\pi)|^p]\,\Gamma(\mathrm{d}s) \to 0 \quad \text{as} \quad h \downarrow 0 \tag{B.3}$$

where $\Gamma(\mathrm{d}s) := (\gamma^{s-t}/C_{T,t,\gamma})\,\mathrm{d}s$ for $C_{T,t,\gamma} := (\gamma^{T-t} - 1)/\log\gamma$. We now prove Theorem 3.5.

**Theorem 3.5.** *If $r$ and $f$ are bounded, then $\lim_{h\downarrow 0} W_p(\zeta_h^\pi(t,x,a), \eta^\pi(t,x)) = 0$, for all $(t,x,a) \in$* $\mathsf{T} \times \mathsf{X} \times \mathsf{A}$*; hence,* $\lim_{h\downarrow 0} \mathsf{distgap}_p(\zeta_h^\pi, t, x) = 0$.

*Proof.* Observe that

$$
\begin{aligned}
\mathsf{distgap}_p(\zeta_h^\pi, t, x) &= \min_{a_1 \neq a_2} W_p(\zeta_h^\pi(t,x,a_1), \zeta_h^\pi(t,x,a_1)) \\
&\leq \min_{a_1 \neq a_2} W_p(\zeta_h^\pi(t,x,a_1), \eta^\pi(t,x)) + \min_{a_1 \neq a_2} W_p(\zeta_h^\pi(t,x,a_2), \eta^\pi(t,x)) \\
&\leq 2 \max_a W_p(\zeta_h^\pi(t,x,a), \eta^\pi(t,x)).
\end{aligned}
$$

Thus, as claimed, it suffices to show that

$$
W_p(\zeta_h^\pi(t,x,a), \eta^\pi(t,x)) \to 0 \quad \text{as} \quad h \downarrow 0,
$$

for all $(t,x,a) \in \mathsf{T} \times \mathsf{X} \times \mathsf{A}$. By (B.1), it suffices to show that

$$
\mathbf{E}[|Z_h^\pi(t,x,a) - G^\pi(t,x)|^p] \to 0 \quad \text{as} \quad h \downarrow 0, \tag{B.4}
$$

for all $(t,x,a) \in \mathsf{T} \times \mathsf{X} \times \mathsf{A}$.

Since $(s,\omega) \mapsto X_s^\bullet(\omega)$ is measurable for $\bullet \in \{\pi, \pi|_{h,a,t}\}$, by Jensen's inequality and Fubini's theorem, we see that

$$
\begin{aligned}
\mathbf{E}[|Z_h^\pi(t,x,a) - G^\pi(t,x)|^p] &\leq 2^{p-1} C_{T,t,\gamma}^p \int_t^T \mathbf{E}[|r(s, X_s^{\pi|_{h,a,t}}) - r(s, X_s^\pi)|^p]\, \Gamma(\mathrm{d}s) \\
&\quad + 2^{p-1}\gamma^{p(T-t)}\mathbf{E}[|(f(X_T^{\pi|_{h,a,t}}) - f(X_T^\pi))|^p].
\end{aligned} \tag{B.5}
$$

In turn, (B.4) follows from (B.2) and (B.3). $\qquad\square$

If $T < \infty$ and $h < 1$, the inequalities in the statements of Lemmas B.2 and B.3 yield the following inequality: for all $p \in [1, \infty)$,

$$
\mathbf{E}[|X_s^{\pi|_{h,a,t}} - X_s^\pi|^p] \leq C_{(\text{B.6})} h^{p/2} \quad \forall s \in \mathsf{T}, \tag{B.6}
$$

for some finite positive constant $C_{(\text{B.6})}$ depending on $p$, $|x|$, $t$, $T$, $C_{2.2}$, and $C_{2.3}$. With this inequality in hand, we now restate and prove Theorem 3.7.

**Theorem 3.7.** *If $r$ is Lipschitz in state, uniformly in time, $f$ is Lipschitz, and $T < \infty$, then $W_p(\zeta_h^\pi(t,x,a), \eta^\pi(t,x)) \lesssim h^{1/2}$, for all $(t,x,a) \in \mathsf{T} \times \mathsf{X} \times \mathsf{A}$; hence, $\mathsf{distgap}_p(\zeta_h^\pi, t, x) \lesssim h^{1/2}$.*

*Proof.* Arguing as in the proof of Theorem 3.5, we see that

$$
\mathbf{E}[|Z_h^\pi(t,x,a) - G^\pi(t,x)|^p] \leq 2^{p-1} C_{T,t,\gamma}^p \mathrm{I} + 2^{p-1}\gamma^{p(T-t)}\mathrm{II}
$$

with

$$
\mathrm{I} := \int_t^T \mathbf{E}[|r(s, X_s^{\pi|_{h,a,t}}) - r(s, X_s^\pi)|^p]\, \Gamma(\mathrm{d}s) \quad \text{and} \quad \mathrm{II} := \mathbf{E}[|(f(X_T^{\pi|_{h,a,t}}) - f(X_T^\pi))|^p].
$$

This is (B.5). In turn, by (B.6) and that $r$ is Lipschitz in space, uniformly in time and $f$ is Lipschitz, we deduce that

$$
\mathrm{I} + \mathrm{II} \leq C_{(\text{B.6})}(C_r^p + C_f^p) h^{p/2},
$$

as desired, where $C_r$ and $C_f$ are the Lipschitz constants of $r$ and $f$ respectively. $\qquad\square$

Recall $r : \mathsf{T} \times \mathsf{X} \to \mathbb{R}$ is Lipschitz in state, uniformly in time if a finite positive constant $C_r$ exists such that

$$
\sup_t |r(t,x) - r(t,y)| \leq C_r |x - y| \quad \forall x, y \in \mathsf{X}.
$$

As promised, we now prove Theorem 3.6.

**Theorem 3.6.** *MDPs and policies exist in and under which, for all $(t,x,a) \in \mathsf{T} \times \mathsf{X} \times \mathsf{A}$, we have that $W_p(\zeta_h^\pi(t,x,a), \eta^\pi(t,x)) \gtrsim h^{1/2}$ and $\mathsf{distgap}_p(\zeta_h^\pi, t, x) \gtrsim h^{1/2}$.*

*Proof.* Let $\mathsf{X} = \mathbb{R}$ and $\mathsf{A} = \{0, 1\}$; set $b \equiv 0$ and $\sigma = \mathbf{1}_{\{a=1\}}$; for all $(s, y) \in \mathsf{T} \times \mathsf{X}$, let $r(s, y) = y$; and set $f \equiv 0$. In words, our action space has two elements, and when action 1 is executed, the system follows Brownian dynamics, and otherwise, the state is fixed. Now consider the policy $\pi$ which always selects the action 0: $\pi(\cdot \,|\, s, y) = \delta_0$, for all $(s, y) \in \mathsf{T} \times \mathsf{X}$.

**Case 1: $\gamma = 1$ and $T < \infty$.** Observe that

$$Z_h^\pi(t, x, 1) - Z_h^\pi(t, x, 0) =_{\text{law}} \int_0^h \tilde{B}_s \,\mathrm{d}s + (T - t - h)\tilde{B}_h,$$

where $(\tilde{B}_s)_{s \geq 0}$ is a Brownian motion. Hence, $Z_h^\pi(t, x, 1) - Z_h^\pi(t, x, 0)$ is equal in law to the sum of two zero mean Gaussian random variables with variances $\sigma_1^2 = h^3/3$ and $\sigma_2^2 = (T - t - h)^2 h$ respectively. And so, it is also Gaussian, and its variance is $\sigma_h^2 = \sigma_1^2 + \sigma_2^2 + 2c\sigma_1\sigma_2$ for some $-1 \leq c \leq 1$. In particular, $\sigma_h^p \gtrsim h^{p/2}$. Recall that central absolute $p$th moment of a Gaussian random variable is proportional to its standard deviation to the power $p$, for all $p \geq 1$. Therefore, by (B.1), we deduce that $\mathsf{distgap}_p(\zeta_h^\pi, t, x) \gtrsim h^{1/2}$, for $h < 1$, as desired.

**Case 2: $T \in [0, \infty]$ and $\gamma \in (0, 1)$.** Observe that

$$Z_h^\pi(t, x, 1) - Z_h^\pi(t, x, 0) =_{\text{law}} N(h) + \frac{\gamma^{T-t} - \gamma^h}{\log \gamma} \tilde{B}_h \quad \text{with} \quad N(h) := \int_0^h \gamma^s \tilde{B}_s \,\mathrm{d}s,$$

for some Brownian motion $(\tilde{B}_s)_{s \geq 0}$. We claim that $N(h)$ is a mean zero Gaussian random variable with variance $\sigma_h^2 \approx h^3/3$. Then the concluding argument in the proof of Case 1 also concludes the proof of this case.

To prove our claim, first note that

$$\mathbf{E}[N(h)] = \int_0^h \gamma^s \mathbf{E}[\tilde{B}_s] \,\mathrm{d}s = 0 \quad \text{and} \quad \mathbf{E}[N(h)^2] = \int_0^h \int_0^h \gamma^{s+s'} \mathbf{E}[\tilde{B}_s \tilde{B}_{s'}] \,\mathrm{d}s\mathrm{d}s'.$$

As

$$\gamma^{2h} \frac{h^3}{3} = \gamma^{2h} \int_0^h \int_0^h \min\{s, s'\} \,\mathrm{d}s\mathrm{d}s' \leq \mathbf{E}[N(h)^2] \leq \int_0^h \int_0^h \min\{s, s'\} \,\mathrm{d}s\mathrm{d}s' \leq \frac{h^3}{3},$$

we see that $N(h)$ has the claimed statistics. Second, observe that $N(h) = \lim_{n \to \infty} N_n(h)$ where

$$N_n(h) := \lim_{n \to \infty} \sum_{i=0}^{n-1} \gamma^{s_i} \tilde{B}_{s_i} d_{h,n} \quad \text{with} \quad d_{h,n} := \frac{h}{n} \text{ and } s_i := id_{h,n}.$$

(This is simply a Riemann sum approximation of the integral that defines $N(h)$.) As the sum of any finite number of Gaussian random variables is a Gaussian random variable, $N_n(h)$ is Gaussian. Furthermore, as the limit of a sequence of Gaussian random variables whose sequences of means and variances converge (to finite values) is Gaussian, $N(h)$ is Gaussian, which proves our claim. $\qquad\square$

## B.2 Distributional Superiority

In this section, we prove the statements and claims made in Section 4.

First, we prove that the mean of every $\psi_h^\pi(t, x, a) \in \mathscr{D}(\zeta_h^\pi(t, x, a), \eta^\pi(t, x))$ is $Q_h^\pi(t, x, a) - V^\pi(t, x)$.

**Lemma B.4.** *If* $\psi_h^\pi(t, x, a) \in \mathscr{D}(\zeta_h^\pi(t, x, a), \eta^\pi(t, x))$, *then its mean is* $Q_h^\pi(t, x, a) - V^\pi(t, x)$.

*Proof.* If $\kappa_h^\pi(t, x, a)$ be such that $\Delta_{\#}\kappa_h^\pi(t, x, a) = \psi_h^\pi(t, x, a)$, then

$$\int r \,\psi_h^\pi(t, x, a)(\mathrm{d}r) = \int (z - w) \,\kappa_h^\pi(t, x, a)(\mathrm{d}z\mathrm{d}w)$$

$$= \int z \,\zeta_h^\pi(t, x, a)(\mathrm{d}z) - \int w \,\eta^\pi(t, x)(\mathrm{d}w)$$

$$= Q_h^\pi(t, x, a) - V^\pi(t, x),$$

as desired. $\qquad\square$

Second, we prove Theorem 4.3.

**Theorem 4.3.** *Let $\kappa \in \mathscr{C}(\mu, \mu)$ for some $\mu \in \mathscr{P}(\mathbb{R})$. The push-forward of $\kappa$ by $\Delta$ is the delta at zero, $\Delta_{\#}\kappa = \delta_0$, if and only if $\kappa$ is a $W_p$-optimal coupling, for some $p \in [1, \infty)$. Moreover, there is only one such coupling. It is given by $\kappa_\mu := (\mathrm{id}, \mathrm{id})_{\#}\mu$ or, equivalently, $\kappa_\mu := (F_\mu^{-1}, F_\mu^{-1})_{\#}\mathcal{U}(0, 1)$. Here $\mathcal{U}(0, 1)$ is the uniform distribution on $[0, 1]$.*

First, we establish that there is only one $W_p$ optimal coupling between a given $\mu \in \mathscr{P}(\mathbb{R})$ and itself, for every $p \in [1, \infty)$.

**Lemma B.5.** *Let $\mu \in \mathscr{P}(\mathbb{R})$. There is only one $W_p$ optimal coupling between $\mu$ and itself, for every $p \in [1, \infty)$. It is $\kappa_\mu := (\mathrm{id}, \mathrm{id})_{\#}\mu \in \mathscr{C}(\mu, \mu)$.*

*Proof.* Let $\kappa \in \mathscr{C}(\mu, \mu)$, and suppose there exists $\epsilon > 0$ for which

$$c_\epsilon := \kappa(\{|z - w| \geq \epsilon \,:\, z, w \in \mathbb{R}\}) > 0.$$

Then

$$\int |z - w|^p \, \kappa(\mathrm{d}z\mathrm{d}w) \geq \int_{\{|z-w| \geq \epsilon\}} |z - w|^p \, \kappa(\mathrm{d}z\mathrm{d}w) \geq |\epsilon|^p c_\epsilon > 0 = \int |z - w|^p \, \kappa_\mu(\mathrm{d}z\mathrm{d}w).$$

Hence, $\kappa$, as considered, is not optimal. Since $\epsilon$ was arbitrary, it follows that a $W_p$ optimal coupling is concentrated on $\{z = w\}$. Therefore, every optimal coupling is of the form $(\mathrm{id}, \mathrm{id})_{\#}\nu$ for some $\nu \in \mathscr{P}(\mathbb{R})$. As the marginals of such a coupling are $\nu$ and $\nu$, we deduce that $\nu = \mu$, as desired. $\square$

*Proof of Theorem 4.3.* Let $\kappa \in \mathscr{C}(\mu, \mu)$ be such that $\Delta_{\#}\kappa = \delta_0$. Observe that

$$\int |z - w|^p \, \kappa(\mathrm{d}z\mathrm{d}w) = \int r^2 \, \delta_0(\mathrm{d}r) = 0 = \int |z - w|^p \, \kappa_\mu(\mathrm{d}z\mathrm{d}w),$$

where, again, $\kappa_\mu = (\mathrm{id}, \mathrm{id})_{\#}\mu$. Hence, $\kappa = \kappa_\mu$, by Lemma B.5. On the other hand, since $\kappa_\mu$ is concentrated on $\{z = w\}$, for any bounded, continuous function $g$, we see

$$\int g(r) \, \Delta_{\#}\kappa_\mu(\mathrm{d}r) = \int g(z - w) \, \kappa_\mu(\mathrm{d}z\mathrm{d}w) = \int_{\{z=w\}} g(z - w) \, \kappa_\mu(\mathrm{d}z\mathrm{d}w) = g(0).$$

In turn, $\Delta_{\#}\kappa_\mu = \delta_0$, as desired. $\square$

**Remark B.6.** *Under the hypotheses of Theorem 3.7, we see that the $1/2$-rescaled superiority distributions at any $(t, x, a)$ for $h \in (0, 1]$ is a family of probability measures with uniformly bounded second moment. Hence, this family is tight. So, up to subsequences, these rescalings converges to limiting probability measure as $h \downarrow 0$. An interesting open question, is whether or not these subsequential limits are the same.*

Third, we prove Theorems 4.5 and 4.6. The proofs of these theorems are a consequence of the following expression for the $W_p$ distance between $\mu$ and $\nu$ when $\mu, \nu \in \mathscr{P}(\mathbb{R})$:

$$W_p(\mu, \nu)^p = \int_0^1 |F_\mu^{-1}(\tau) - F_\nu^{-1}(\tau)|^p \, \mathrm{d}\tau.$$

**Theorem 4.5.** *MDPs and policies exist satisfying Assumptions 2.2 and 2.3 in and under which, for all $(t, x) \in \mathsf{T} \times \mathsf{X}$, we have that $\mathsf{distgap}_p(\psi_{h;q}^\pi, t, x) \gtrsim h^{1/2-q}$.*

*Proof.* Recall that $F_{\psi_{h;q}^\pi(t,x,a)}^{-1} = h^{-q}(F_{\zeta_h^\pi(t,x,a)}^{-1} - F_{\eta^\pi(t,x)}^{-1})$, for all $a \in \mathsf{A}$. Hence, for every $a_1 \neq a_2$,

$$W_p(\psi_{h;q}^\pi(t, x, a_1), \psi_{h;q}^\pi(t, x, a_2))^p = \int_0^1 |F_{\psi_{h;q}^\pi(t,x,a_1)}^{-1}(\tau) - F_{\psi_{h;q}^\pi(t,x,a_2)}^{-1}(\tau)|^p \, \mathrm{d}\tau$$

$$= h^{-qp} \int_0^1 |F_{\zeta_h^\pi(t,x,a_1)}^{-1}(\tau) - F_{\zeta_h^\pi(t,x,a_2)}^{-1}(\tau)|^p$$

$$= h^{-qp} W_p(\zeta_h^\pi(t, x, a_1), \zeta_h^\pi(t, x, a_2))^p.$$

Thus, taking the $p$th root of both sides of the above equality, we deduce that

$$W_p(\psi_{h;q}^\pi(t,x,a_1), \psi_{h;q}^\pi(t,x,a_2)) = h^{-q}W_p(\zeta_h^\pi(t,x,a_1), \zeta_h^\pi(t,x,a_2)).$$

The example presented in Theorem 3.6 is such that $W_p(\zeta_h^\pi(t,x,a_1), \zeta_h^\pi(t,x,a_2)) \gtrsim h^{1/2}$. Whence, $W_p(\psi_{h;q}^\pi(t,x,a_1), \psi_{h;q}^\pi(t,x,a_2)) \gtrsim h^{1/2-q}$. $\qquad\square$

**Theorem 4.6.** *Under Assumptions 2.2 and 2.3, if $r$ is Lipschitz in state, uniformly in time, $f$ is Lipschitz, and $T < \infty$, then $\mathsf{distgap}_p(\psi_{h;q}^\pi, t, x) \lesssim h^{1/2-q}$, for all $(t,x) \in \mathsf{T} \times \mathsf{X}$.*

The proof of Theorem 4.6 is almost identical to the proof of Theorem 4.5.

*Proof.* Arguing as in the proof of Theorem 4.5, we see that

$$W_p(\psi_{h;q}^\pi(t,x,a_1), \psi_{h;q}^\pi(t,x,a_2)) = h^{-q}W_p(\zeta_h^\pi(t,x,a_1), \zeta_h^\pi(t,x,a_2)).$$

By Theorem 3.7, it follows that $W_p(\psi_{h;q}^\pi(t,x,a_1), \psi_{h;q}^\pi(t,x,a_2)) \lesssim h^{1/2-q}$. $\qquad\square$

Finally, we prove Theorem 4.8

**Theorem 4.8.** *Let $\rho_\beta$ be a distortion risk measure, $q \geq 0$, and $h > 0$. If $\rho_\beta(\eta^\pi(t,x)) < \infty$, then $\arg\max_{a \in \mathsf{A}} \rho_\beta(\psi_{h;q}^\pi(t,x,a)) = \arg\max_{a \in \mathsf{A}} \rho_\beta(\zeta_h^\pi(t,x,a))$.*

*Proof.* Observe that

$$
\begin{aligned}
\rho_\beta(\psi_{h;q}^\pi(t,x,a)) &= \int_0^1 F^{-1}_{\psi_{h;q}^\pi(t,x,a)}(\tau)\,\beta(\mathrm{d}\tau) \\
&= \int_0^1 h^{-q} F^{-1}_{\psi_h^\pi(t,x,a)}(\tau)\,\beta(\mathrm{d}\tau) \\
&= h^{-q} \int_0^1 (F^{-1}_{\zeta_h^\pi(t,x,a)}(\tau) - F^{-1}_{\eta^\pi(t,x)}(\tau))\,\beta(\mathrm{d}\tau) \\
&= h^{-q}\rho_\beta(\zeta_h^\pi(t,x,a)) - h^{-q}\rho_\beta(\eta^\pi(t,x))
\end{aligned}
$$

Since $\rho_\beta(\eta^\pi(t,x))$ is independent of $a$ and $h > 0$,

$$\arg\max_a \rho_\beta(\psi_{h;q}^\pi(t,x,a)) = \arg\max_a \rho_\beta(\zeta_h^\pi(t,x,a)).$$

$\qquad\square$

## C   Algorithms and Pseudocode

Here we discuss methods for policy optimization via distributional superiority. In practice, computers operate at a finite frequency—as such, all policies we consider here will be assumed to apply each action for $h$ units of time, as in the settings of [34, 16].

Before describing the superiority learning algorithms, we first remark on the form of exploration policies used in our approaches. We consider policies of the form

$$\pi^{\mathrm{explore}} : \mathbb{R}^\mathsf{A} \times \mathbb{R}^n \to \mathsf{A}.$$

The first argument to such a policy is a vector of "action-values". In our case, given a superiority distribution $\psi_{h;q}^\pi(t,x,\cdot)$, the action values may be $(\rho_\beta(\psi_{h;q}^\pi(t,x,a)))_{a \in \mathsf{A}}$ for a distortion risk measure $\rho_\beta$. This generalizes the notion of $Q$-values for distributional learning.

The second argument represents a noise variable, in order to support stochastic policies. As input to our algorithms, we require a probability measure $\mathbb{P}_{\mathrm{act}}$ on processes $(\epsilon_t)_{t \geq 0}$. This framework generalizes common exploration methods in deep RL. For example, to recover $\epsilon$-greedy policies, $\mathbb{P}_{\mathrm{act}}$ represent a 2-dimensional white noise, and

$$\pi^{\mathrm{explore}}(v,\xi) = \begin{cases} \arg\max_a v_a & \text{if } \xi_1 \leq \epsilon \\ a_{\lceil \xi_2 |\mathsf{A}| \rceil} & \text{otherwise.} \end{cases} \tag{C.1}$$

Alternatively, one might choose to correlate the action noise. Approaches such as DDPG [21] and DAU present such examples, where action noise evolves over time as an Ornstein-Uhlenbeck process. This can be implemented in the framework above by choosing $\mathbb{P}_{\text{act}}$ as the distribution of an $|A|$-dimensional Ornstein-Uhlenbeck process, and defining

$$\pi^{\text{explore}}(v, \xi) = \arg\max_{a \in A} (v + \xi). \tag{C.2}$$

In our experiments, we found that $\epsilon$-greedy exploration was sufficient for approximate policy optimization. To keep our implementation closest to the QR-DQN baseline, therefore, our implementations use the definition of $\pi^{\text{explore}}$ from equation (C.1).

The remainder of this section details the implementation of DSUP($q$) and DAU+DSUP($q$) as introduced in Section 4.2. Additionally, we provide source code for our implementations at https://github.com/harwiltz/distributional-superiority.

### C.1 Distributional Superiority

Generally, our goal is to learn a $\rho_\beta$-greedy policy (see Definition 4.7). Since all policies apply actions for $h$ units of time, in order to satisfy Axiom 2, we want

$$S_h^\pi(t, x, \pi(x)) \equiv 0,$$

where $\pi : x \mapsto \arg\max_a \rho_\beta(S_h^\pi(t, x, a))$ is the $\rho_\beta$-greedy policy. As such, following the DAU algorithm of [34], our algorithms will model quantile functions $\phi(t, x, a) \in \mathbb{R}^m$ that aim to satisfy

$$F^{-1}_{\psi^\pi_{h;q}(t,x,a;h)} \approx \phi(t, x, a) - \phi(t, x, a^\star) \quad \text{for some} \quad a^\star \in \arg\max_a \rho_\beta(\phi(t, x, a)).$$

Our primary algorithmic contribution integrates such a model, with proper superiority rescaling, into the QR-DQN framework of [10] for estimating action-conditioned return distributions. It is outlined in Algorithm 2.

To deal with the increased time-resolution of transitions, [34] modified the step size by a factor of $h$. More recently, [3] found that subsampling transitions before storing them in the replay buffer was most effective in their high-decision-frequency domain. Thus, we opt for such a strategy here. Rather than only storing every $h^{-1}$ transitions, we randomly select transitions to store according to independent Bernoulli($h$) draws in order to avoid the possibility of only capturing cyclic phenomena in the replay buffer. We found that this strategy worked similarly to that of [34], but is far less computationally expensive. Likewise, as $h$ decreases, we extend the number of training interactions by a factor of $h^{-1}$. This corresponds to training for a constant amount of time units across decision frequencies, and likewise, a constant number of gradient updates across decision frequencies.

### C.2 Two-Timescale Advantage-Shifted Distributional Superiority

An astute reader might recognize that while $\psi^\pi_{h;q}$ may have nonzero distributional action gap, since $h^q$ is asymptotically larger than $h$ for $q < 1$, the works of [34, 16] would suggest that the *expected* action gap under $\psi^\pi_{h;q}$ should vanish. To account for this, we propose shifting the rescaled superiority quantiles by the advantage function, as estimated e.g. in DAU [34]. It is clear that such a procedure cannot cause the distributional action gap to vanish. The resulting procedure is depicted in Algorithm 3, with the modifications relative to Algorithm 2 highlighted in blue. In practice, we employ a shared feature extractor in the representations of $A_h$ and $\phi$ to reap the representation learning benefits of DRL [4] when approximating the advantage.

### C.3 Influence of the Rescaling Factor

The algorithms 2 and 3 are parameterized by a *rescaling factor* $q \in (0, 1]$, which is meant to compensate for the collapse of the distributional action gap. Larger values of $q$ correspond to larger compensation. In this work, we argue that the distributional action gap collapses at rate $h^{1/2}$, leading to a natural choice of $q = 1/2$, which theoretically preserves constant order action gaps with respect to the decision frequency. We also test the approach with $q = 1$, which corresponds to the well-known scaling rate for preserving expected value action gaps.

For any $q > 1/2$, the distributional action gap theoretically grows without bound as $h \downarrow 0$, leading to distributional estimates with arbitrarily large variance. On the other hand, for any $q < 1/2$, the distributional action gap decays to 0 as $h \downarrow 0$, which makes it difficult to identify the best actions in the presence of approximation error.

**Algorithm 2** Distributional Superiority Quantile Regression

---

**Require:** $m$ (number of quantiles) and $q$ (rescale factor)
**Require:** $h^{-1}$ (decision frequency), $\mu$ (initial state distribution)
**Require:** $\pi^{\text{explore}}$ (exploration policy) and $\rho_\beta$ (distortion risk measure)
**Require:** $\mathbb{P}_{\text{act}}$ (action noise distribution)
**Require:** $\theta(t,x), \phi(t,x,a) \in \mathbb{R}^m$, parameterized quantile representations for $\eta^\pi(t,x), \psi^\pi_{h;q}(t,x,a)$.
**Require:** $\bar{\theta}(t,x) \in \mathbb{R}^m$, target quantile representations for $\eta^\pi(t,x)$.
**Require:** $n_\theta$ (target update period), $\alpha$ (step size), $N$ (batch size)

  $\mathcal{D} \leftarrow \emptyset$
  `reset` $\leftarrow$ `True`
  **for** $i \in \mathbb{N}$ **do**
    // Gather data with exploratory policy
    $(\xi)_{s \geq 0} \sim \mathbb{P}_{\text{act}}$                  $\triangleright$ Sample from stochastic process for exploration noise
    **for** $j \in \{0, \ldots, \lfloor h^{-1} \rfloor\}$ **do**
      **if** `reset` is `True` **then**
        $t \leftarrow 0$ and $x_0 \sim \mu$
        `reset` $\leftarrow$ `False`
      **end if**
      $a^\star \leftarrow \arg\max_a \rho_\beta \left( \frac{1}{m} \sum_{n=1}^m \delta_{\phi(t,x_t,a)_n} \right)$
      $F^{-1}_{\psi_{h;q}}(a) \leftarrow \phi(t,x_t,a) - \phi(t,x_t,a^\star)$ for each $a \in \mathsf{A}$
      $v_a \leftarrow \rho_\beta \left( \frac{1}{m} \sum_{n=1}^m \delta_{F^{-1}_{\psi_{h;q}}(a)_n} \right)$ for each $a \in \mathsf{A}$
      $a_t \sim \pi^{\text{explore}}((v_a)_{a \in \mathsf{A}}, \xi_{jh})$
      Perform action $a_t$ for $h$ units of time, observe $(r_t, x_{t+h}, \mathsf{done}_{t+h})$.
      $Y_j \sim \mathsf{Bernoulli}(h)$              $\triangleright$ Subsample transitions for replay
      **if** $Y_j = 1 \lor \mathsf{done}_{j+1} = \mathsf{True}$ **then**
        $\mathcal{D} \leftarrow \mathcal{D} \cup \{(t, x_t, a_t, r_t, x_{t+h}, \mathsf{done}_{t+h})\}$
      **end if**
      `reset` $\leftarrow \mathsf{done}_{t+h}$ and $t \leftarrow t + h$
    **end for**

    // Update return/superiority distributions
    Sample minibatch $\{(t_k, x_k, a_k, r_k, x'_k, \mathsf{done}_k)\}_{k=1}^N$ from $\mathcal{D}$
    **for** $k \in [N]$ **do**
      $a^\star_k \leftarrow \arg\max_a \rho_\beta \left( \frac{1}{m} \sum_{n=1}^m \delta_{\phi(t_k,x_k,a)_n} \right)$      $\triangleright$ Risk-sensitive greedy action
      $F^{-1}_\zeta(\phi, \theta) \leftarrow \theta(t_k, x_k) + h^q(\phi(t_k, x_k, a_k) - \phi(t_k, x_k, a^\star))$      $\triangleright$ Prediction (4.4)
      $\mathcal{T}F^{-1}_\zeta \leftarrow hr_k + \gamma^h(1 - \mathsf{done}_k)\bar{\theta}(t_k + h, x'_k) + \gamma^h\mathsf{done}_k f(x'_k)$      $\triangleright$ Target (4.5)
      $\ell_k(\phi, \theta) \leftarrow \mathsf{QuantileHuber}(F^{-1}_\zeta(\phi, \theta), \mathcal{T}F^{-1}_\zeta)$      $\triangleright$ See [10, Eq. (10)]
    **end for**
    $\ell(\phi, \theta) \leftarrow \frac{1}{N} \sum_{k=1}^N \ell_k(\phi, \theta)$
    $\phi \leftarrow \phi - \alpha\nabla_\phi \ell(\phi, \theta)$ and $\theta \leftarrow \theta - \alpha\nabla_\theta \ell(\phi, \theta)$      $\triangleright$ Gradient updates
    **if** $i \mid n_\theta$ **then**
      $\bar{\theta} \leftarrow \theta$
    **end if**
  **end for**

---

**Algorithm 3** Advantage-Shifted Distributional Superiority Quantile Regression

**Require:** $m$ (number of quantiles) and $q$ (rescale factor)
**Require:** $h^{-1}$ (decision frequency), $\mu$ (initial state distribution)
**Require:** $\pi^{\text{explore}}$ (exploration policy) and $\rho_\beta$ (distortion risk measure)
**Require:** $\mathbb{P}_{\text{act}}$ (action noise distribution)
**Require:** $\theta(t,x), \phi(t,x,a) \in \mathbb{R}^m$, parameterized quantile representations for $\eta^\pi(t,x), \psi^\pi_{h;q}(t,x,a)$.
**Require:** $\overline{\theta}(t,x) \in \mathbb{R}^m$, target quantile representations for $\eta^\pi(t,x)$.
**Require:** $\widetilde{A}_h(t,x,a)$ (parameterized representation of advantage function)
**Require:** $n_\theta$ (target update period), $\alpha$ (step size), $N$ (batch size)

  $\mathcal{D} \leftarrow \emptyset$
  reset $\leftarrow$ True
  **for** $i \in \mathbb{N}$ **do**
    *// Gather data with exploratory policy*
    $(\xi)_{s \geq 0} \sim \mathbb{P}_{\text{act}}$                          $\triangleright$ Sample from stochastic process for exploration noise
    **for** $j \in \{0, \ldots, \lfloor h^{-1} \rfloor\}$ **do**
      **if** reset is True **then**
        $t \leftarrow 0$ and $x_0 \sim \mu$
        reset $\leftarrow$ False
      **end if**
      $a^\star \leftarrow \arg\max_a \rho_\beta \left( \frac{1}{m} \sum_{n=1}^m \delta_{\phi(t,x_t,a)_n + (1-h^{1-q})\widetilde{A}_h(t,x_t,a)} \right)$
      $F^{-1}_{\psi_{h;q}}(a) \leftarrow \phi(t,x_t,a) - \phi(t,x_t,a^\star)$ for each $a \in \mathsf{A}$
      $A_h(t,x_t,\cdot) \leftarrow \widetilde{A}_h(t,x_t,\cdot) - \widetilde{A}_h(t,x_t,a^\star)$                     $\triangleright$ see [34, DAU]
      $v_a \leftarrow \rho_\beta \left( \frac{1}{m} \sum_{n=1}^m \delta_{F^{-1}_{\psi_{h;q}}(a)_n + (1-h^{1-q})\widetilde{A}_h(t,x_t,a)} \right)$ for each $a \in \mathsf{A}$
      $a_t \sim \pi^{\text{explore}}((v_a)_{a \in \mathsf{A}}, \xi_{jh})$
      Perform action $a_t$ for $h$ units of time, observe $(r_t, x_{t+h}, \text{done}_{t+h})$.
      $Y_j \sim \text{Bernoulli}(h)$                                  $\triangleright$ Subsample transitions for replay
      **if** $Y_j = 1 \vee \text{done}_{j+1} = \text{True}$ **then**
        $\mathcal{D} \leftarrow \mathcal{D} \cup \{(t, x_t, a_t, r_t, x_{t+h}, \text{done}_{t+h})\}$
      **end if**
      reset $\leftarrow \text{done}_{t+h}$ and $t \leftarrow t + h$
    **end for**

    *// Update return/superiority distributions*
    Sample minibatch $\{(t_k, x_k, a_k, r_k, x'_k, \text{done}_k)\}_{k=1}^N$ from $\mathcal{D}$
    **for** $k \in [N]$ **do**
      $a^\star \leftarrow \arg\max_a \rho_\beta \left( \frac{1}{m} \sum_{n=1}^m \delta_{\phi(t_k,x_k,a)_n + (1-h^{1-q})\widetilde{A}_h(t_k,x_k,a)} \right)$

      *// Advantage Loss*
      $A_h(t_k,x_k,a_k) \leftarrow \widetilde{A}_h(t_k,x_k,a_k) - \widetilde{A}_h(t_k,x_k,a^\star)$
      $V(t_k,a_k) \leftarrow \frac{1}{m} \sum_{n=1}^{\cdot} \theta(t_k,x_k)_n$
      $Q(t_k,x_k,a_k) \leftarrow V(t_k,a_k) + hA_h(t_k,x_k,a_k)$
      $\mathcal{T}Q(t_k,x_k,a_k) = hr_k + \gamma^h \frac{1}{m} \sum_{n=1}^m \overline{\theta}(t_k+h, x_{k+1})_n$
      $\ell^{\text{adv}}_k(\widetilde{A}_h) \leftarrow (Q(t_k,x_k,a_k) - \mathcal{T}Q(t_k,x_k,a_k))^2$                         $\triangleright$ Bellman error

      *// Superiority Loss*
      $F^{-1}_\zeta(\phi,\theta) \leftarrow \theta(t_k,x_k) + h^q(\phi(t_k,x_k,a_k) - \phi(t_k,x_k,a^\star))$         $\triangleright$ Prediction (4.4)
      $\mathcal{T}F^{-1}_\zeta \leftarrow hr_k + \gamma^h(1-\text{done}_k)\overline{\theta}(t_k+h, x'_k) + \gamma^h \text{done}_k f(x'_k)$         $\triangleright$ Target (4.5)
      $\ell_k(\phi,\theta) \leftarrow \text{QuantileHuber}(F^{-1}_\zeta(\phi,\theta), \mathcal{T}F^{-1}_\zeta)$               $\triangleright$ See [10, Eq. (10)]
    **end for**
    $\ell(\phi,\theta) \leftarrow \frac{1}{N} \sum_{k=1}^N \ell_k(\phi,\theta)$ and $\ell^{\text{adv}}(\widetilde{A}_h) \leftarrow \frac{1}{2N} \sum_{k=1}^N \ell^{\text{adv}}_k(\widetilde{A}_h)$
    $\phi \leftarrow \phi - \alpha \nabla_\phi \ell(\phi,\theta)$ and $\theta \leftarrow \theta - \alpha \nabla_\theta \ell(\phi,\theta)$ and $\widetilde{A}_h \leftarrow \widetilde{A}_h - \alpha \nabla \ell^{\text{adv}}(\widetilde{A}_h)$
    **if** $i \mid n_\theta$ **then**
      $\overline{\theta} \leftarrow \theta$
    **end if**
  **end for**

# D  Additional Experimental Results

Figure D.1 include results comparing the performance of DSUP(1/2) and QR-DQN for risk-sensitive option trading across a variety of decision frequencies.

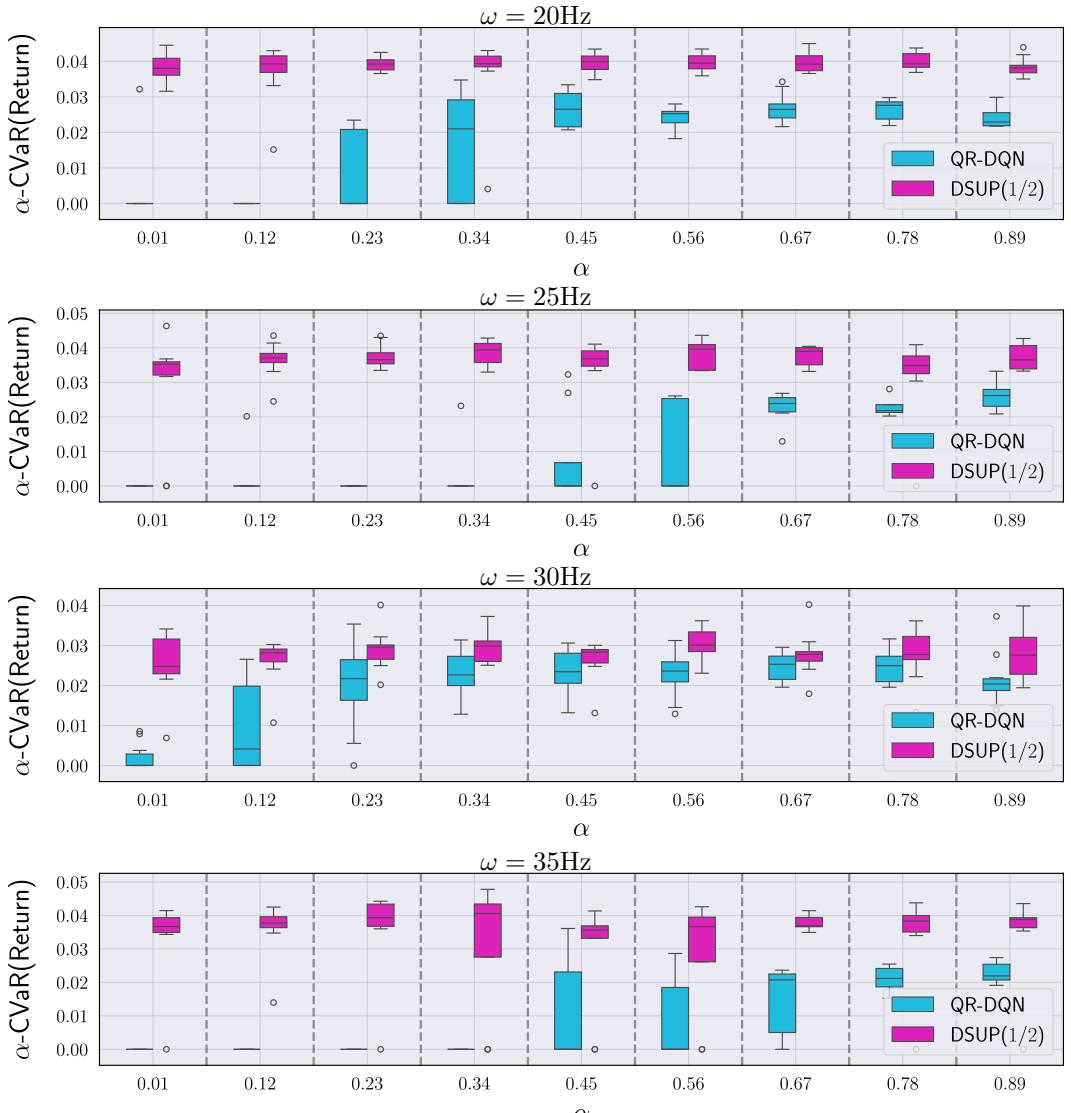

Figure D.1: Risk-sensitive option trading performance for various decision frequencies $\omega$.

# E  Simulation Details

Here we collect further information about the setup for the simulations described in Section 5.2.

## E.1  Option Trading Environment

The environment used for the high-frequency option-trading setup is identical to that of Lim and Malik [22]. The environment emulates policies that decide when to exercise American call options. The state space is modeled as $X = \mathbb{R}$ and with $T = [0, T]$. Notably, existing works such as [22] and [17] describe the state space by $X = \mathbb{R}^2$, where one dimension represents time—in our setup, we generally condition policies and returns on time, so we indeed model policies and returns as functions on $\mathbb{R}^2$. The state $X_t$ represents the price of the option at time $t$, and evolves according to a geometric Brownian motion.

There are two actions in the environment. Action 0 "holds" the option, while action 1 represents "execute". Upon taking action 1 (or equivalently, once the time reaches $T$), the option is *executed*, and the agent receives a reward $f(x) = \max(0, 1-x)$ and the episode terminates; here $x$ represents the price at the time of execution. No rewards are incurred otherwise.

Following the setup of [22], the dynamics of the prices are simulated based on data collected between years 2016 and 2019 from 10 commodities on the DOW market. Lim and Malik proposed a method for estimating the most likely parameters of geometric Brownian motion to fit the data for each commodity, which is then used to simulate many environment rollouts for training and evaluation. This is particularly convenient for our setup, where we additionally scale the decision frequency, corresponding to finer time discretizations of the Euler-Maruyama scheme for the estimated geometric Brownian motion. Like [22], separate dynamics parameters are estimated (for each commodity) between training and evaluation: the dynamics used for training are estimated on prefixes of the data, and those for testing (post-training) are estimated on suffixes of the data. Results are reported on the testing dynamics, averaged over the 10 commodities.

As is standard [22, 17], we simulate the environment with $T = 100$ and $X_0 = 1$. The simulations from [22] correspond to $h = 1$ in this setting. In our high-frequency simulations, we discretize the dynamics with timestep $h < 1$.

### E.2 Hyperparameters

In Table 1, we list the hyperparameters used in the simulations for the tested algorithms. We note that although the original DAU implementation scaled the learning rate with $h$, we take an alternative approach by updating every $h^{-1}$ environment steps akin to [3]. This is discussed in more detail in Appendix C.

Table 1: Hyperparameters

| Method | Parameter | Value |
|---|---|---|
| All | Replay buffer sampling | Uniform |
| | Replay buffer capacity | 20000 |
| | Batch size | 32 |
| | Optimizer | Adam [18] |
| | Learning rate | 1e-4 |
| | Discount factor | 0.999 |
| | Target network update period | 1000 |
| | $\epsilon$-greedy exploration parameter | Linear decay from 1 to 0.02 |
| DAU [34] | Value network | MLP (2 hidden, 100 units, ReLU) |
| | Advantage network | MLP (2 hidden, 100 units, ReLU) |
| QR-DQN [10, 22] | Quantile network torso | MLP (2 hidden, 100 units, ReLU) |
| | Number of atoms | 100 |
| | Quantile Huber parameter $\kappa$ | 1 |
| DSUP | $\eta$ network torso | MLP (2 hidden, 100 units, ReLU) |
| | $\phi$ (superiority proxy) network torso | MLP (2 hidden, 100 units, ReLU) |
| | Number of atoms | 100 |
| | Quantile Huber parameter $\kappa$ | 1 |
| DAU+DSUP | Advantage network torso | (shared with $\phi$) |

### E.3 Compute Resources

Our implementations are written in Jax [6] and executed with a single NVidia V100 GPU. At highest decision frequencies, experiments took longer to execute, averaging out at a maximum of roughly four hours.

