# OpenReview forum: "Action Gaps and Advantages in Continuous-Time Distributional Reinforcement Learning"
_NeurIPS.cc/2024/Conference — NeurIPS 2024 poster_

### Official Review · Reviewer_dYRz · 2024-07-11

**Soundness:** 3
**Presentation:** 3
**Contribution:** 3
**Rating:** 6
**Confidence:** 2

**Summary:**

This paper focuses on the scenario of continuous RL with high decision frequency. They show that it is hard for distributional RL agents to accurately estimate action values when the decision frequency increases, just like in the case of ordinary RL agents. They propose a distributional analogy of the action gap in ordinary RL called superiority. Based on this notion they try to mitigate this issue by the rescaling technique and propose a superiority-based algorithm. They also validate the proposed methods by numerical simulations.

**Strengths:**

* The paper is overall well-written and easy to follow.

* The idea of superiority distributions is interesting and intuitive, with solid theoretical justifications.

* The work is quite complete. Besides the discussion about the action gap in distributional RL and the notion of superiority distributions, the author also proposed new algorithms and performed extensive empirical studies.

Therefore, I choose to give a positive rating of this work. However, given that I am unfamiliar with continuous RL, I choose to assign a low confidence score to my review.

**Weaknesses:**

* It seems that the basic idea of the superiority distribution is that for two distributions $\eta_1,\eta_2$, we define an object $\Delta$ that can be viewed as the difference between $\eta_1$ and $\eta_2$. And then given a risk-sensitive objective $\phi$, one may choose actions according to rescaled $\phi(\Delta/h^q)$, where $h$ denotes the decision frequency. But why not simply use the criterion $(\phi(\eta_1)-\phi(\eta_2))/h^q$ for decision making? I think the authors should discuss this issue in later versions of this paper.

* I suggest the author add some explanations about the problem setting to the intro section, which would make the paper more readable for those not familiar with the specific field.

**Questions:**

* I note that the superior distributions are actually rescaled by $h^{-q}$, where $q$ is a tuning parameter. In the empirical studies, $q$ is set to be $1$ or $1/2$. Are there any possible heuristic rules for the choice of $q$?

* It seems that in the simulation studies, the performances of some methods (DAU, DSUP(1), DAU+DSUP(1/2)) are not monotone w.r.t. the decision frequency, but oscillate a lot as the decision frequency increases. Why would this happen? Can the authors give a more detailed explanation?

**Limitations:**

It seems the authors do not explicitly addressed the limitations of their work.

---

> ### Author Rebuttal · Authors · 2024-08-07
>
> We thank the reviewer for their thorough assessment of our work, their interest in its results, and insightful comments.
>
> **Re: Discussion on using $(\phi(\eta_1) - \phi(\eta_2))/h^q$ for decision making.**
>
> Thank you for pointing this out. We appreciate your suggestion here. We will be happy to add text that speaks to this in our revised draft. In practice, it is generally difficult to estimate $(\phi(\eta_1) - \phi(\eta_2))$ without estimating $\eta_1$ and $\eta_2$ (see, e.g., Chapter 7 of *Distributional Reinforcement Learning*, Bellemare, Dabney, and Rowland 2023). This is one reason why distributional RL is so attractive. Moreover, there are other benefits to modeling return distributions. For instance, modeling distributions is helpful in obtaining second order regret bounds (see *More Benefits of Being Distributional: Second Order Bounds for Reinforcement Learning* by Wang et al., 2024). Finally, even if we could estimate $(\phi(\eta_1) - \phi(\eta_2))$ directly, some form of rescaling would still be necessary. This is one of the novel contributions of our work.
>
> **Re: The problem setting.**
>
> Thank you for raising our awareness to this. We very much appreciate this suggestion, and we will be sure to update our revision with this in mind. We discussed this in further detail in our general response.
>
> **Q1**: Tuning $q$.
>
> **A1**: In reality, $q$ isn’t a hyperparameter to tune. The theoretical section of our work shows that the only appropriate/principled choice of $q$ is ½. For all other choices, distributional action gaps either blow up or vanish, as $h$ decreases. Both of these behaviors are undesirable. We model $q = 1$ for benchmarking purposes, since this corresponds to rescaling in AU/DAU. That said, if one insists on treating $q$ as a hyperparameter, choosing larger $q$ will further increase the distributional action gap, but at the cost of extremely large variance; this will likely hinder performance, as seen by the performance of DSUP(1), for example.
>
> **Q2**: Oscillatory performance of DAU, DSUP(1), DAU+DSUP(½).
>
> **A2**: This is an important point to clarify, thank you for making it! Please see the general response.
>
> **Re Limitations**:
>
> As noted in our checklist, limitations are addressed throughout the text where relevant. For instance, note all technical restrictions stated formally as Assumptions. The only other restriction is on the class of reward functions, which we specify in Footnote 1. These Assumptions and Footnote 1, while stated in full precision in the main text, are elaborated on in Appendix A.
>
> If there is something explicit you would like us to address, please let us know; we are happy to do so.

---

> > ### Comment · Reviewer_dYRz · 2024-08-12
> >
> > I want to thank the authors for the detailed response. However, I think the authors may have misunderstood my point on the difference between considering $\phi(\eta_1) - \phi(\eta_2)$ and $\phi(\Delta/h^q)$. Here the idea is not to directly estimate $\phi(\eta_1) - \phi(\eta_2)$ but to first estimate $\eta_1$ and $\eta_2$ (with existing distributional RL techniques) and then make decisions according to $\phi(\eta_1) - \phi(\eta_2)$. I think this method is conceptually simpler than the methodology proposed in the paper. I hope the authors can provide reasons why they choose to adopt the (seemingly more complexi) proposed methodology.

---

> > > ### Author Response · Authors · 2024-08-12
> > >
> > > Thanks for clearing that up! Indeed we did misinterpret the original question.
> > >
> > > The issue with your suggestion is that it still involves learning $\eta_1$ itself. This is what we want to avoid: as $h$ decreases, the distributions $\{\eta_1(x, a)\}_{a\in\mathcal{A}}$ will be all roughly equal in the $W_p$ metrics (Theorems 3.5 and 3.7) – the distributional action gap is small. Therefore, $\arg\max_a(\phi(\eta_1(x, a)) - \phi(\eta_2(x)))$ is going to be highly corrupted by approximation error.
> > >
> > > Your suggestion is actually equivalent to the QR-DQN baseline in Figure 5.4 (and Figure 5.2 for the risk-neutral case). Note that, in your example, $\eta_2$ is a function of state only, so it doesn’t affect the ranking of actions. Likewise, scaling $\phi(\eta_1(x, a))$ or $(\phi(\eta_1(x, a)) - \phi(\eta_2(x)))$ by $h^{-q}$ will not change the ranking of actions. Estimating $\eta_1$ and acting according to $\phi(\eta_1(x, a))$ (which is the same as acting according $\phi(\eta_1(x, a)) - \phi(\eta_2(x))$) is precisely what the QR-DQN baseline is doing in Figure 5.4, and we see that this struggles (again, due to the vanishing distributional action gap).
> > >
> > > Our method circumvents this by learning the rescaled superiority directly. That is, rather than modeling $\eta_1$ and $\eta_2$ and constructing $\Delta/h^{q}$ from those, we directly model $\Delta/h^{q}$. For instance, see equation 4.1 or Algorithm 1: we never model $\zeta(x, a)$ (which is $\eta_1$ in your example), we only model the rescaled superiority $\psi_{h;1/2}(x, a)$. This preserves the distributional action gap as shown by Theorem 4.8 (and Figure 5.3 for an empirical demonstration), preserves the correct ranking of actions as shown by Theorem 4.10, and performs much better empirically as a consequence, as shown in Figures 5.2 and 5.4.

---

> > > > ### Comment · Reviewer_dYRz · 2024-08-13
> > > >
> > > > Thanks for the response. All of my major concerns are properly addressed. So I decide to keep my positive rating of this paper.

---

### Official Review · Reviewer_ZgER · 2024-07-11

**Soundness:** 3
**Presentation:** 3
**Contribution:** 3
**Rating:** 6
**Confidence:** 3

**Summary:**

This paper investigates the action gap in distributional RL, where the decisions are made at high frequency. The authors showed that the distributional RL is also sensitive to the decision frequency. In particular, they proved the action-conditioned return distribution collapse with different rates for the statistics. Also, they introduce the generalization of the advantage function in continuous-time MDP, i.e., superiority, based on which a novel algorithm is designed. The authors finally conducted extensive experiments in the high-frequency option-trading domain to demonstrate their theory and the efficacy of their algorithms.

**Strengths:**

* The writing is rigorous with detailed and clear definitions and theorems.

* Algorithmic practice in this setting is creditable.

* Experiments are relatively extensive, considering both illustrated and comparative settings.

**Weaknesses:**

* The motivation can be strengthened. What is an action gap in continuous-time MDP, and what is an algorithm's outcome when encountering high-frequency decision-making settings? Why do we investigate the distributional RL in this setting? More explanation would be better.

* The clarity can be enhanced. Although I appreciate the rigorous writing, it would be better to emphasize the main conclusions of this paper. For instance, the statistics in distributional RL also decrease at different rates. How do readers learn from these observations? Some rigorous theorems may be put in the appendix and replaced with more straightforward explanations.

**Questions:**

1. While the authors stated that DAU+DSUP(1/2) is inferior to DSUP(1/2), it is indeed a clear drawback of the proposed algorithm as this empirical result is consistent with the theory stated in Section 4.2, and hard to understand to me. Although the authors gave some explanations, they did not rigorously validate their hypothesis. Also, I am not convinced why DSUP(1/2) performs poorly in low-frequency domains.

2. Why only consider 30HZ and DSUP(1/2) in the risk-sensitive setting? My suggestion is to include DAU+DSUP(1/2) as well for an extensive comparison across different frequencies, which could be more convincing.

3. It is suggested that more examples be given to emphasize the significance of studying continuous-time MDP and the motivation to explore distributional RL.

4. It would be better to provide the pointers to the proof of each theorem to allow readers to check the proof instantly.

5. In what sense does the approximation hold in Eq. 4.2?

**Limitations:**

yes

---

> ### Author Rebuttal · Authors · 2024-08-07
>
> We thank the reviewer for their thorough assessment of our work, their interest in its results, and insightful comments.
>
> **Re: Motivation and Clarity.**
>
> Thank you for pointing to these potential areas of improvement. We very much appreciate your suggestions. We will be sure to update our revision with this in mind. Please see the general response for some more detail on how we will do this.
>
> **Q1a**: DAU+DSUP(½) vs DSUP(½).
>
> **A1a**: This is an important point to clarify. Thank you for making it! Please see the general response.
>
> **Q1b**: DSUP(½) at low decision frequency.
>
> **A1b**: We do not agree that DSUP(½) performs poorly at low decision frequencies. Figure 5.2 shows that DSUP(½)’s performance is similar to the best performer in the lowest two frequencies (the only two where it is not the best performer). These lowest frequencies are the only ones where the baselines are competitive, since action gaps are not an issue here.
>
> **Q2a**: Other frequencies in CVaR experiment.
>
> **A2a**: Among the higher decision frequencies, we felt that QR-DQN stood a fighting chance only in the 30Hz setting. This choice was based on how significantly DSUP(½) out-performs QR-DQN (see Figure 5.2). We have since run the experiment at several decision frequencies as you suggested. A PDF of these results appears with the general response document. We felt they would be of interest to all the reviewers. Note that the trend observed in Figure 5.4 persists across the full range of now-considered decision frequencies.
>
> **Q2b**: DAU+DSUP(½) in CVaR experiment.
>
> **A2b**: We note that DAU+DSUP(½) *is not* principled for risk-sensitive control. In particular, DAU+DSUP(½) does not preserve the ranking of actions with respect to CVaR relative to the action-conditioned return distributions (i.e., Theorem 4.10 only holds for DAU+DSUP(½) when $\beta$ is the uniform measure on $[0,1]$, that is, when we are in the risk-neutral or expected-value setting). However, Theorem 4.10 shows that DSUP(½) *is* principled for general $\beta$.
>
> **Q3**: Additional motivating examples.
>
> **A3**: We will be sure to point to more motivating examples in our revised draft.
>
> **Q4**: Pointers to proofs.
>
> **A4**: We will hyperlink the proofs directly beneath the theorems in our revision.
>
> **Q5**: Approximation in Eq. 4.2.
>
> A5: The approximation holds in the sense that the support of the law of the error $Y_h$ is contained within a $o(h)$ radius of $0$. Please see the exact equation just above line 247.

---

> > ### Comment · Reviewer_ZgER · 2024-08-11
> >
> > Thanks for the authors' response. Although the paper can be further enhanced by improving the motivation and strengthening the experiments, I believe it is an interesting supplement to distributional RL literature in terms of continuous time and has great potential. Thus, I keep the positive assessment.

---

### Official Review · Reviewer_TDv5 · 2024-07-13

**Soundness:** 4
**Presentation:** 3
**Contribution:** 3
**Rating:** 7
**Confidence:** 2

**Summary:**

The paper investigates the issues around continuous-time distributional reinforcement learning. In traditional RL, the advantage becomes less informative as the frequency of actions increases, vanishing at the limit and making it impossible to distinguish between actions. This work extends this result to distributional RL in several ways: it first proposes a framework based on Wasserstein distances between return distributions, then proves in this framework that a similar problem appears in DLR, and gives tight bounds on the asymptotic convergence rate of the distgap (an action gap analogue for continuous DLR), establishing that the rate differs from the traditional RL case.

After that, the paper constructs a notion of superiority - a DLR analogue of advantage, and considers its different variants (transformations) that ensure that action gaps are preserved in the limit. Finally, two families of algorithms are constructed, and shown to outperform the baseline QR-DQN on a benchmark of high-frequency options trading task.

**Strengths:**

The paper investigates a very interesting question, and gives a theoretically satisfactory answers. The formal analysis gives rise to a practical algorithm, which is empirically shown to outperform the QR-DQN baseline. The presentation is very readable while being maximally mathematically precise, which I really appreciated.

**Weaknesses:**

The theoretical part requires quite a lot of background from the reader. While authors give a short intro to SDEs in the appendix, it is, in my opinion, nowhere near the depth required to understand the paper. While it's not really possible to present a comprehensive intro to stochastic analysis, the paper would benefit, in my opinion, from a slight expansion of the relevant appendix. Also in terms of presentation, I found the explanation of the algorithmic part of the paper really dense and confusing, and I cannot say I understood it particularly well from the main text.

**Questions:**

1. The Wasserstein distances considered are for $p \in [1, \infty)$. What breaks if we put $p = \infty$?
2. Why is the behavior of $\vartheta$ mean non-monotone wrt frequency in the example considered in sec 5.1?
3. Apart form distributional RL, could a similar method be applied to continuous-time max-ent RL?

**Limitations:**

Yes.

---

> ### Author Rebuttal · Authors · 2024-08-07
>
> We thank the reviewer for their thorough assessment of our work, their interest in its results, and insightful comments.
>
> **Re: Background on SDEs.**
>
> Thank you for pointing this out. We appreciate this suggestion. We will happily expand our background on SDEs. We plan to incorporate discussion on and references to the relationship between MDPs governed by SDEs and MDPs governed by transition probabilities, as they are presented more traditionally in RL. We feel this might make readers primarily familiar with discrete-time RL, for instance, more comfortable with the setting of the theory part of our paper as well as some of our technical assumptions.
>
> Is there something else explicit you feel would benefit our work?
>
> **Re: Density of Algorithm Section.**
>
> Again, thank you for pointing this out. We appreciate this as well. We will happily expand this section too. In our revision, we will add detailed descriptions (in Section 4.2) and pseudocode (in Appendix C) for both DAU and QR-DQN. We feel this should help clarify how our proposed methods deviate and model the rescaled superiority distribution.
>
> Do you have specific suggestions that add to what we have proposed to incorporate?
>
> **Q1**: $p = \infty$.
>
> **A1**: Theorems 3.7 and 4.8, for example, break in the case of $p = \infty$. That said, to freely work with $W_\infty$, our state processes should have bounded sample paths, which is violated even in the case of Brownian sample paths. (See, e.g., Section 5.5.1 of *Optimal Transport for Applied Mathematicians* by Santambrogio for more details on working with $W_\infty$.) However, since $W_\infty \geq W_p$ for all $p$, some of our analysis does hold: Proposition 3.4, Theorem 3.6, Theorem 4.5, and Theorem 4.7.
>
> **Q2**: Non-monotonicity of the mean of $\vartheta^\pi_{h;1/2}$ in Section 5.1.
>
> **A2**: Good question, thank you for pointing this out! The true distributions should actually all have the same mean, roughly 100. They do not because of Monte Carlo approximation error (MCAE). Furthermore, as we are operating at extremely high decision frequency, errors are amplified quite a bit. This is less apparent in the other subplots because the variance blow-up and/or mean collapse dominates the MCAE.
>
> **Q3**: Continuous-time MaxEnt RL.
>
> **A3**: This is a really interesting question! Yes, our theory provides insight into the distributional properties of returns (and the issues with their estimation from data) in this setting, as a function of the decision frequency, because MaxEnt RL can be seen as an instance of expected-value RL with a policy-dependent reward function. Thus, we believe our DSUP algorithms would be effective for learning return distributions in continuous-time MaxEnt RL as well.

---

> > ### Comment · Reviewer_TDv5 · 2024-08-12
> > **Response**
> >
> > I thank the authors for their thorough response. Re: specific additional suggestions: I agree with the other reviewers, that adding heuristic, natural language explanations would benefit the paper. At the same time, given the space constraints, I understand it might be difficult to maintain the level of the mathematical rigour, which I value more highly.
> >
> > When reading the paper, I looked at the Algorithm 1 in the Appendix C, but could not follow the details - I would recommend either splitting it into sub-procedures, or otherwise giving a higher-level abstraction description, because right now it is a page-long wall-of-text, which makes comprehending it a quite daunting task.
> >
> > I decided to keep my original (positive) rating.

---

### Author Rebuttal · Authors · 2024-08-07

# General Response

We are thankful for the interest in our work as well as the time and effort taken to review it.

Reviewers praised our work for the problem that we have highlighted and the completeness of our theoretical treatment of it (TDv5, ZgER, dYRz), our general clarity and rigor of exposition (TDv5, dYRz), and our extensive empirical evaluation (TDv5, ZgER, dYRz). In their assessments, the reviewers suggested our paper would benefit from more heuristic discussion of the problem setting, asked for clarity regarding the behavior of some of the algorithms as a function of decision frequency, and requested for our risk-sensitive experiments to be conducted over additional decision frequencies.

We speak to these suggestions and requests here; other queries posed by reviewers are addressed in our individual responses to them.

**Problem Setting: Motivation and Clarity**

Reviewers ZgER and dYRz suggested our paper would benefit from more heuristic discussion of the problem setting and motivations for its study to aid in its accessibility.

We appreciate this feedback, and we will amend our introduction accordingly. We propose to preface the introduction with content regarding the scope of the problem setting along the lines of the following text.

“Our work investigates the performance of RL agents in systems where states evolve continuously in time, but policies make decisions at discrete time steps ($h$ units of time apart). Many real-world deployments of RL are within such systems, since the world is continuous in time, yet computerized RL policies operate in discrete time. Within these systems, many factors influence the decision frequency, such as the quality of sensors and the speed of CPUs. And while more responsive policies should perform better, this is not always true in practice. As such, our goal is to design algorithms that perform well across the continuum of decision frequencies.”

Furthermore, in our revised introduction, we will be sure to use the prompts given by ZgER as guides to further expand the accessibility of our work.

**Performance of DAU, DSUP(1), and DAU+DSUP(½)**

Reviewers ZgER and dYRz were curious about the performance of DAU, DSUP(1), and DAU+DSUP(½).

The issue here is purely with respect to estimating $h$-rescaled advantages in stochastic environments: the $h$-rescaled superiority, as we showed theoretically and via simulation, has unbounded variance as $h$ decreases. Thus, estimating its mean (the $h$-rescaled advantage) is challenging and sensitive to noise when $h$ is small. Our work is the first, to our knowledge, to highlight this issue.

We see the oscillatory performance of DAU, DSUP(1), and DAU+DSUP(½) as additional symptoms of this issue. Indeed, their individual performances track one another, because they all use $h$-rescaled advantage estimates for control. In DAU and DAU+DSUP(½), this is done explicitly. In DSUP(1), it is done implicitly (by modeling the $h$-rescaled superiority distribution and acting according to its expectation).

Overall, our work is the first (to our knowledge) to highlight the statistical challenges of modeling $h$-rescaled advantages in stochastic environments. Our main practical focus was to design an algorithm that maintains distributional action gaps, which is accomplished by DSUP(½). Further investigation into improved methods for modeling $h$-rescaled advantages in stochastic environments is an important direction we leave to future work.

In our revised draft, we will expand our presentation of DAU, DSUP(1), and DAU+DSUP(½) to contain what we have briefly outlined here.

**Risk-Sensitive Control Across Decision Frequencies**

Reviewer ZgER suggested that our analysis of the performance of our risk-sensitive superiority-based algorithms can be strengthened by testing across more decision frequencies.

We include some further results to this end in the attached PDF. In particular, we test DSUP(½) and QR-DQN at 3 additional frequencies, to round out the 4 highest frequencies plotted in Figure 5.1. Note that DSUP(½) out-performs QR-DQN in these 3 additional decision frequencies as well. Also, these results confirm our expectations from our theoretical results: DSUP(½) is stable across decision frequencies, while QR-DQN struggles at these high decision frequencies (even more so than we expected).

---

> ### Author Response · Authors · 2024-08-10
>
> Thank you again for your time and thoughts. We’re keen to discuss our work with you, especially in light of your insightful comments. If you have any remaining questions, we would be happy to address them; please let us know.

---

### Decision · Program_Chairs · 2024-09-25

**Decision:**

Accept (poster)

**Comment:**

This paper investigates the challenges arising from dealing with continuous-time control in distributional RL. In particular, whenever decision are made at a high frequency, it comes harder to compare the relative advatage of actions, thus making it difficult or impossible to distinguish between them. This paper introduces a framework exploiting Wasserstein distances between return distributions to provide bounds on the convergence rate of the gap between actions and generalizes the notion of advantage functions to continuous-time MDPs.

All reviewers had an overall positive view of this work, highlighting the importance of the problem being tackled by the authors and the thoroughness of the presented theory. ZgER praised the relatively extensive experiments conducted by the authors. Reviewer ZgER suggested that the empirical analyses presented in this work be strengthened by testing the proposed idea across additional decision frequencies, which the authors did. Reviewers brought up a few (relatively minor) points of contention, and the discussion phase was fruitful: the authors did address the most important concerns mentioned in the reviews and pointed out concrete ways in which their work can and will be improved in terms of presentation and clarity.

The most common and pressing concern regarding this paper was its clarity. Although reviewers agreed that the writing is rigorous, the consensus is that the paper is not necessarily accessible to a broad audience due to the highly dense text and the authors not always discussing intuitions that underlie the ideas being investigated and proposed. This will most likely limit how accessible the contributions introduced in this paper are to the NeurIPS community. The reviewers strongly encouraged the authors to improve their document by adding further intuition and discussion and by improving clarity to ensure that the insights introduced in this work become more apparent to a broader audience of readers who may not be, a priori, experts on all theoretical concepts exploited in this paper.